



# Nonlinear response of tropical lower stratospheric temperature and water vapor to ENSO

Chaim I Garfinkel[1], Amit Gordon[1], Luke D Oman[2], Feng Li[3], Sean Davis[4], and Steven Pawson[2]

[1]The Fredy and Nadine Herrmann Institute of Earth Sciences, Hebrew University of Jerusalem, Jerusalem, Israel.
[2] NASA Goddard Space Flight Center, Greenbelt, MD, USA.
[3] Universities Space Research Association, Columbia, MD, USA.
[4] NOAA Earth System Research Laboratory, Boulder, CO, USA.

*Correspondence to:* Chaim I. Garfinkel (chaim.garfinkel@mail.huji.ac.il)

**Abstract.** A series of simulations using the NASA Goddard Earth Observing System Chemistry-Climate Model are analyzed in order to assess interannual and sub-decadal variability in tropical lower stratospheric temperature and water vapor over the past 35 years. The impact of El Niño-Southern Oscillation in this region is nonlinear. While moderate El Niño events lead to cooling in this region, strong El Niño events appear to lead to warming, even as the response of the large scale Brewer Dobson Circulation appears to scale nearly linearly with El Niño. The tropospheric warming associated with strong El Niño events extends into the tropical tropopause layer and up to the cold point, where it allows for more water vapor to enter the stratosphere. The net effect is that both strong La Niña and strong El Niño events lead to enhanced entry water vapor and stratospheric moistening. These results lead to the following interpretation of the millennial drop in water vapor in 2001: the very strong El Niño event in 1997/1998 followed by more than two consecutive years of La Niña led to enhanced lower stratospheric water vapor. As this period ended in early 2001, entry water vapor concentrations declined. The net effect is that sea surface temperature variability led to a decrease in water vapor of 0.14ppmv after 2001, which accounts for approximately 23% of the observed drop.

## 1 Introduction

The El Niño - Southern Oscillation (ENSO) is the largest source of interannual variability in the Tropics, and manifests as anomalous sea surface temperatures in the Eastern and Central Pacific Ocean. El Niño (EN), the phase with anomalously warm sea surface temperatures in this region, has been shown to impact stratospheric temperatures in both the polar region and in the Tropics (Calvo Fernández et al., 2004; Sassi et al., 2004; Manzini et al., 2006; Garcia-Herrera et al., 2006; Taguchi and Hartmann, 2006; Garfinkel and Hartmann, 2007; Marsh and Garcia, 2007; Free and Seidel, 2009; Calvo et al., 2010). The temperature response in these regions is linked, as ENSO is able to modify the stratospheric mean meridional circulation, also known as the Brewer-Dobson (BD) circulation. During a EN event, anomalous upward propagation and dissipation of planetary waves at middle and high latitudes, and gravity waves and transient synoptic waves in the subtropics (Garfinkel and Hartmann, 2008; Calvo et al., 2010; Simpson et al., 2011), leads to the acceleration of the BD circulation, resulting in a cooler tropical lower stratosphere and warmer polar stratosphere.



In addition to impacting zonal mean tropical lower stratospheric temperatures, ENSO also impacts the zonal distribution of temperature anomalies. EN leads to a Rossby wave response whereby anomalously warm temperatures are present over the Indo-Pacific warm pool (hereafter warm pool) near the tropopause, with colder temperatures further east over the Central Pacific (Yulaeva and Wallace, 1994; Randel et al., 2000; Zhou et al., 2001; Scherllin-Pirscher et al., 2012). In the tropical tropopause
layer water vapor increases in the region with warm anomalies and decreases in the region with cold anomalies, and these local changes in tropical water vapor can exceed 25% below the cold point (Gettelman et al., 2001; Konopka et al., 2016).

The net effect of these temperature anomalies on water vapor above the tropical cold point is complex, as these zonally asymmetric changes are superposed on the larger scale warming or cooling associated with changes of the BDC. The two largest EN events in the satellite era (in 1997/1998 and in 2015/2016) clearly preceded moistening of the tropical lower
stratosphere (Fueglistaler and Haynes, 2005; Avery et al., 2017), though the impact of more moderate events is less clear. The net effect of EN in water vapor at the cold point is the residual of the large temperature anomalies in the West Pacific and Central Pacific (Gettelman et al., 2001; Davis et al., 2013; Konopka et al., 2016), and zonally averaged changes in entry water vapor for the moderately-strong ENSO events considered by Gettelman et al. (2001) is 0.1ppmv. In addition, Calvo et al. (2010), Garfinkel et al. (2013a), and Konopka et al. (2016) note the strong seasonal dependence of the effect of EN on stratospheric
water vapor: only in spring does EN lead to enhanced water vapor and LN to dehydration, and Garfinkel et al. (2013a) argue that this occurs only as the coldest region near the cold-point tropopause moves closer to India and therefore samples the region which warms moreso than the region which cools.

A clearer understanding of the role of ENSO for entry water vapor may be important for understanding the 2000/2001 drop in water vapor (Randel et al., 2004, 2006): Brinkop et al. (2016) argue that the evolution of ENSO from 1997 through 2000 was
crucial for this event, such that ENSO variability aliased onto sub-decadal variability. As the amount of water vapor that enters the stratosphere is important for stratospheric chemistry (Solomon et al., 1986) and radiative balance (Forster and Shine, 1999; Solomon et al., 2010), it is important to understand the factors that control its entry into the stratosphere on all timescales.

This paper is motivated by three specific issues related to the lower stratospheric response to ENSO: First, a commonly used method to ascribe stratospheric variability to forcings such as ENSO, the QBO, solar variability, and volcanoes, is to
use multiple linear regression (e.g. Crooks and Gray, 2005; Marsh and Garcia, 2007; Mitchell et al., 2015). An assumption underlying this method is that the response to these forcings is linear, i.e. that the response to a given magnitude El Niño is equal and opposite to that of a La Niña event of equal magnitude. Is this assumption really true? Second, Garfinkel et al. (2013a) found that EN events whose sea surface temperature anomalies peak in the Central Pacific (i.e. CP events) lead to dehydration regardless of season while events peaking in the Eastern Pacific (i.e. EP events) lead to spring moistening. However, EP
events tend to be stronger than CP events, and it is not clear to what extent the difference found by Garfinkel et al. (2013a) reflects the intensity of the EN event or the flavor of the event. Finally, it has been suggested that SST variability in the Pacific Ocean contributed to the post-2000 drop in water vapor (Rosenlof and Reid, 2008; Garfinkel et al., 2013b) possibly via ENSO (Brinkop et al., 2016), but this contribution has not yet been quantified.

This paper will demonstrate that there are nonlinearities in the lower stratospheric response to ENSO. While typical EN
events lead to tropical lower stratospheric cooling and dehydration in winter, the springtime response is nonlinear: strong EN



events and LN lead to moistening while weak/moderate EN events lead to dehydration. We clarify that discriminating between CP and EP events may not be crucial, and rather one should discriminate between very strong EN events and moderate EN events. As CP events tend to be weaker than East Pacific events (Johnson, 2013), it is easy to confuse a composite of CP EN events with a composite of moderate EN regardless of type. Finally, we suggest that SST variability in the early 2000s relative to the late 1990s led to 0.14ppmv of dehydration, explaining approximately 23% of the drop in water vapor over this period (Randel et al., 2004, 2006).

The data and methods are introduced in Sections 2 and 3. Section 4 demonstrates the nonlinearity of ENSO's effect on tropical lower stratospheric temperature and water vapor. In order to better understand the nonlinearities evident in Section 4, Section 5 considers more closely the strongest EN event covered by our model experiments: the event in the winter of 1997/1998. Section 6 considers implications of the interannual variability for the millennial drop in 2001 and for the EN event in 2015/2016. The appendix (Section 8) discusses the linearity of the influence of ENSO on the BDC.

## 2 Data

We analyze the MERRA (Modern-era retrospective analysis for research and applications; Rienecker et al., 2011) reanalysis, the merged water vapor product from SWOOSH v2.5 (Davis et al., 2016), and output from atmospheric chemistry-climate general circulation models (GCMs) and coupled ocean-atmosphere GCMs on various time scales. The Goddard Earth Observing System Chemistry-Climate Model, Version 2 (GEOSCCM Rienecker et al, 2008)) couples the GEOS-5 (Rienecker et al, 2008; Molod et al., 2012) atmospheric general circulation model to the comprehensive stratospheric chemistry module StratChem (Pawson et al., 2008). The model has 72 vertical layers, with a model top at 0.01 hPa, and all simulations discussed here were performed at 2° latitude x 2.5° longitude horizontal resolution. The model spontaneously generates a QBO (Molod et al., 2012). The model vertical levels between 140hPa and 50hPa are located at 139.1hPa, 118.3hPa, 100.5hPa, 85.4hPa, 72.6hPa, 61.5hPa, and 52.0hPa; output is plotted at standard pressure levels.

The convection scheme used in GEOSCCM is based on Relaxed Arakawa-Schubert (Moorthi and Suarez, 1992; Rienecker et al, 2008), and the cloud ice parameterization is described in Molod et al. (2012). Note that there is cloud ice in the version of the model under consideration here up to 85hPa (as is evident in our results below). To the extent that entry water vapor is controlled by large scale temperature patterns and the relatively crude ice parameterization in the current generation of the model, we expect that our model captures the response of water vapor to ENSO. That being said, more advanced treatments of ice clouds are currently under development, and hence similar studies must be performed as models improve.

A series of integrations were performed with the GEOSCCM, and they are listed in Table 1 and described below. They fall into two classes: coupled ocean-atmosphere simulations, and historical-SSTs simulations with an atmospheric chemistry-climate general circulation model (AGCM). Both modeling frameworks have their advantages: coupled ocean-atmosphere simulations allow the model to self-consistently develop SST anomalies and teleconnections without violating energetic constraints, and also allow us to examine the stratospheric response to a wider range of ENSO events than have occurred in the





historical record. On the other hand, simulations forced with observed SSTs can be more easily compared to the observed response to ENSO.

The model configuration for the coupled ocean-atmosphere simulation is described in Li et al. (2016). The ocean model is the Modular Ocean Model version 5 (Griffies et al., 2015) with 50 vertical layers, and the ocean horizontal resolution is about 1° latitude by 1° longitude. We consider the last 240 years of 340 year-long simulation in which GHG and ODS forcings are fixed at 1950 levels.

The foundation of the AGCM ensemble are the simulations discussed by Garfinkel et al. (2015) and Aquila et al. (2016), though several recent integrations have been added as summarized in Table 1. The simulations form a 42 member ensemble of the period from January 1980 to December 2009, though five integrations have been extended to the near-present to cover the strong EN event in 2015/2016. Such an ensemble is valuable as it frames the forced response to EN common to all integrations within the context of stochastic unforced variability unique to each integration. For 13 integrations, the only time-varying forcings are changing SSTs and sea ice; SSTs and sea ice up to November 2006 are taken from the Met Office Hadley center observational database (Rayner et al., 2006) and from the National Climatic Data Center (NCDC) (Reynolds et al., 2002) since then. For 3 additional integrations, GHG concentrations are from observations up to 2005 and from the Representative Concentrations Pathway 4.5 after 2005 (Meinshausen et al., 2011) in addition to time varying SSTs and sea ice. For 19 additional integrations ozone depleting substances also vary as observed. For seven additional integrations these forcings plus volcanic eruptions are included (Aquila et al., 2016); for these seven integrations we discard the winter seasons 1991/1992 and 1992/1993 and the years 1991, 1992, and 1993 from consideration, as the eruption of Mt. Pinatubo had a large impact on the BDC and tropical temperatures in our simulations (Aquila et al., 2016; Garfinkel et al., 2017), and appears to have led to moistening in observational data as well (Fueglistaler, 2012; Dessler et al., 2014). In 1994 the difference in entry water vapor between these seven integrations and the other integrations is less than 0.05ppmv (not shown). Four of these seven integrations also include time varying solar forcing. All simulations considered are summarized in Table 1. These simulations have been performed for various purposes and differ in the forcings included and in the physical parameterizations, but they all include changing SSTs and sea-ice.

GEOSCCM model output is compared to temperatures from MERRA and water vapor from SWOOSH v2.5. Temperatures from MERRA are interpolated to the same 2° latitude x 2.5° longitude degree grid used for the GEOSCCM simulations. In order to isolate the interannual variability, we detrend timeseries for the AGCM simulations and for reanalysis/observations. Finally, as the forcings and model generations differ among these integrations, anomalies are computed with respect to each integration.

The advantage in studying historical changes in water vapor and temperature in free running climate simulations is *not* to form a best estimate of the actual interannual variability; for that purpose, nudged experiments and/or Lagrangian trajectory modeling are far better. Rather, the motivation is four-fold: one, future projections of the temperature and water vapor changes in this region can only be produced by free running climate simulations, and these projections are of limited value if a model's simulation of the past is inconsistent with observational constraints; two, assuming the model is capable of capturing interannual variability, the causes of trends or discontinuities (such as the millennial drop) can be better understood in a framework in





which there is no possibility that changes in the observing or modeling system could have led to these trends or discontinuities; three, large ensembles of a free running model can be produced in order to better isolate the forced response from a single EN event from unrelated internal atmospheric variability not forced by anomalies at the ocean surface; fourth, and relatedly, the observational record is not long enough in order to assess whether the response to ENSO is nonlinear, and thus only by

considering large model ensembles can these nonlinear effects be examined.

## 3   Methods

ENSO events are categorized into four groups similar to Hurwitz et al. (2014): Eastern Pacific (EP) EN, characterized by positive sea surface temperature (SST) anomalies in the Nino-3 region (5S - 5N, 210E-270E), and Central Pacific (CP) EN, characterized by positive SST anomalies in the Nino-4 region (5S-5N, 160E-210E), as well as EP and CP La Niña events,

characterized by negative SST anomalies in the same two regions. ENSO events are identified based on NDJF seasonal mean SST anomalies in the ERSSTv4 dataset (Huang et al., 2015) with a 1981-2010 base period, and the same definition is applied to the coupled ocean-atmosphere simulations. EN and LN events are identified when SST anomalies in the Nino3.4 region exceed 0.5K and -0.5K respectively. EN and LN events are further categorized as follows: EP El Niño events are identified when the Nino-3 anomaly is 0.1 K larger than the corresponding Nino-4 anomaly. Similarly, EP La Niña events are identified

when the Nino3 anomaly is 0.1 K less than the Nino-4 anomaly. CP El Niño and CP La Niña events are identified analogously. All remaining years, either because they are neutral ENSO or because the Nino-3 and Nino-4 anomalies are within 0.1K, are categorized as "other events". The years included in each composite are listed in Table 2.

Most ENSO events peak in the late fall or early winter, and decay by the following spring. Hence, we focus on the response of the lower stratosphere during the period from November through June.

As discussed in the introduction, it is well known that EN forces an intensified BDC. Hence, in considering the response to ENSO, we consider the response without regressing out the influence of the BDC, as regressing out the BDC misrepresents the net impact of ENSO on the lower stratosphere. We have analyzed the water vapor response upon regressing out the influence of the BDC, and the moistening of the stratosphere during El Niño is more pronounced as expected. We consider two alternate diagnostics of the BDC: the tropical diabatic heating rate and the mean age; the main text shows results for tropical diabatic

heating rate, and the appendix shows mean age.

A QBO is spontaneously generated in all simulations considered here. The QBO phase differs among these experiments (i.e. the phase does not match observations), and hence many of the complications that arise due to the QBO (e.g. Liang et al., 2011) are not relevant. We have confirmed that regressing out the influence of the QBO at 50hPa has little impact on our model results. However, we linearly regress out the zonal wind at 50hPa two months prior for figures that compare model output to

observations/reanalysis.

Due to the very slow vertical motions in tropical tropopause layer and relatively faster horizontal motions, entry water vapor is sensitive to the coldest regions in the tropics and not just zonal mean temperatures (i.e. the cold point, Mote et al., 1996; Fueglistaler et al., 2004; Fueglistaler and Haynes, 2005; Oman et al., 2008). We therefore include isotherms corresponding



to the coldest region in the tropics on figures of temperature at 100hpa. The climatological cold point is enclosed with a green contour, and the corresponding contour during EN is enclosed in magenta. Temperature anomalies at 85hPa resemble quantitatively those at 100hPa, and we therefore show 100hPa anomalies only for brevity. For the figure showing the time evolution of the cold point temperature, the cold point temperature is defined by first sorting all temperatures at 100hPa from

5S-5N for each calendar month and integration, then choosing the temperature threshold for the coldest decile of gridpoints in this region. Results are similar if we use e.g. 5% or 15%. Similar thresholds were considered by Zhou et al. (2004), Oman et al. (2008), and Garfinkel et al. (2013a).

## 4   Linearity of the ENSO effect in the tropical lower stratosphere

We now consider the seasonality and linearity of the ENSO effect in the tropical lower stratosphere. Figure 1 shows the response

of temperature, the BDC, and water vapor to ENSO in the coupled ocean-atmosphere run, with the top row for late fall and the bottom row for late spring. Figure 2 is comparable but for the AGCM integrations, and Figure 3 is comparable but for MERRA and SWOOSH data. The slope and uncertainty of the linear least-squares best fit is indicated on each panel, and different colors are used to distinguish CP from EP events.

We begin with temperature changes in winter. EN leads to strong cooling of the tropical lower stratosphere in winter (Figure

1ad, 2ad), while LN leads to warming. This temperature response is consistent, to zeroeth order, with the changes in the BDC associated with ENSO: EN leads to an accelerated BDC while LN leads to a decelerated BDC (Figure 1be and 2be). In winter, the relationship between ENSO and lower stratospheric conditions are linear; that is, the impact of EN and LN events of similar strength is equal and opposite. The magnitude of these effects, as quantified by the best-fit line, appears to be slightly weaker in the AGCM ensemble as compared to the coupled ocean-atmosphere runs, and this could be because of the difference in the

nature of ENSO events or decadal variability.

The right column of Figures 1 and 2 considers changes in water vapor. In both the AGCM and the coupled ocean-atmosphere simulations EN leads to dehydration in winter. Even if we linearly regress out the BDC the slope in Jan/Feb is still suggestive of EN leading to dehydration, though the slope is no longer significantly different from zero (not shown). That EN leads to dehydration in winter is in agreement with Calvo et al. (2010), Garfinkel et al. (2013a), and Konopka et al. (2016), who all note

the strong seasonal dependence of the effect of EN on stratospheric water vapor.

While the relationship between ENSO and lower stratospheric conditions is linear in winter, it is nonlinear for both water vapor and temperature in spring (bottom two rows of Figure 1 and 2). Namely, strong EN events lead to less cooling than what might have been expected given the linear best-fit. Consistent with this, the strongest EN events lead to more moistening that might have been expected based on the linear best-fit line. This is especially evident in figure 1il, where the strongest EN lead

to springtime moistening. The AGCM runs seems to capture this effect, as the 97/98 EN also leads to moistening (the most extreme EN event in figure 2il). This effect is explored further in section 5, where we compare the temperature response to the 97/98 EN to other EN events. More generally, a parabolic (e.g. $H_20 \sim a * EN^2$) fit better describes the springtime relationship




between ENSO and water vapor and between ENSO and lower stratospheric temperature than the linear fit shown in Figure 1 and 2.

The net, annual-averaged effect of ENSO on water vapor is the residual of large cancellation between the effects in midwinter and summer, and the sign of this residual differs between the coupled ocean-atmosphere integration and the AGCM integrations

(not shown).

It does not matter whether the ENSO event is categorized as a CP or EP event, as the red, black, and cyan dots all indicate the same relationship between ENSO and water vapor. However, the strongest EN events tend to be EP in both nature and in the coupled ocean integrations, and hence the compositing approach of Garfinkel et al. (2013a) to characterizing the impact of EP events and CP events can mislead due to the nonlinear effects discussed above and in section 5.

Finally, it is important to note that for all panels in Figure 1 and 2, the model simulated evolution falls within the observational constraints ( Figure 3). Furthermore, the qualitatively different behavior for the 1997/1998 event is evident both for the model experiments and observations. Hence the model reasonably simulates nature. However, it impossible to verify nonlinearities in observational/reanalysis data given the relatively short data record, and error bars on the best-fit slope estimates are large.

## 5    Composite analysis of the 97/98 event as compared to other events

In order to better understand why strong EN events may behave differently from weak ones, we compare the 97/98 event to other EN events in the AGCM GEOSCCM runs. The time evolution of the water vapor anomalies associated with the 97/98 event are shown in Figure 4a, and Figure 4b shows the water vapor anomalies associated with all other EN events. There is clearly a large difference, with only the 97/98 event leading to robust moistening.

Figure 5 and 6 show a map view of changes in temperature at 100hPa for the 97/98 event and for all other EP EN events.

While in all cases there is relative cooling in the Central Pacific and relative warming over the Warm Pool region (consistent with Yulaeva and Wallace, 1994; Randel et al., 2000; Scherllin-Pirscher et al., 2012), there are clear differences in the temperature pattern of importance for water vapor. In the 97/98 event in winter, the zero-line of temperature anomalies is $30°$ further east than for the other EP EN events. This zonal shift is of crucial importance for the cold point region, as the cold region shifts east without shrinking for typical EP EN events (compare the magenta and green isotherms in Figure 6ab) but warms

for the 1997/1998 event (i.e. there are no magenta isotherms in Figure 5 due to warming of the cold point). In spring, there is broad-scale warming over most of the equatorial band for the 97/98 event. A similar effect is seen in the MERRA reanalysis (not shown). The net effect is that in both winter and spring, the 97/98 event led to moistening of the stratosphere relative to other EP EN events.

The changes in tropical temperature in MERRA and in GEOSCCM for the 97/98 event and for other events are summarized

in Figures 7 and 8. These figures show the temperature averaged from 5S to 5N from 300hpa to 50hPa. The overall quadrupole structure is similar to that in Liang et al. (2011) and Garfinkel et al. (2013b), and the model captures the warming pattern in reanalysis. There are clear differences between the changes in 97/98 and those in other EN years: the tropospheric warming is





more pronounced and widespread in 97/98. The net effect is that the cold point region warms in 97/98 but not in the other EN years.

It is important to emphasize that this nonlinearity in the temperature and water vapor response appears to originate from the troposphere. The changes in the BDC appear to be mostly linear in Figures 1, 2, and 3. The wave-driving of the BDC is

not the source of the nonlinearity (see the appendix). In contrast, the response to EN is nonlinear in the tropical tropopause layer. Figure 9 shows the temperature anomalies associated with ENSO for pressure levels from 150hPa to 50hPa. EN leads to a warmer troposphere (e.g. at 150hPa) due to enhanced latent heat release in January and February. This 150hPa warming in January and February for the 1997/1998 event is stronger than one might expect from linearity (i.e. the anomaly lies above the best-fit line, such that the residuals from the best-fit line are all positive), and the nonlinearity is even more pronounced

at 100hPa (Figure 9b). The exceptional nature of the equatorial warming associated with the 1997/1998 event evident in the tropical tropopause layer extends up to 85hPa (Figure 9c). The nonlinearity appears to propagate upwards to 70hPa by March and April (Figure 9i). Only at 50hPa does the linearity of the stratospheric BDC response lead to a linear temperature response (Figure 9ej). Similar results are evident in the coupled ocean-atmosphere integration (Figure 10). The net effect is that the 1997/1998 event led to exceptional warming throughout the tropopause transition layer and at the cold point, and hence led to

enhanced water vapor entering the stratosphere.

Why was the 97/98 El Niño tropospheric warming so distinct from other events? While this was the strongest El Niño over the period considered by this paper, the 1982/1983 El Niño was not much weaker than the 1997/1998 event as measured by the Nino3.4 index, yet the impact of the 1982/1983 on water vapor was qualitatively different. Furthermore, the upper tropospheric warming in the Central and East Pacific sectors for the 1982/1983 and 1997/1998 events (Figure 7) are similar.

This suggests that the Central and East Pacific responses cannot explain the difference in stratospheric response. In contrast, these two events differed quite dramatically in the Indian Ocean (and more generally in zonally averaged tropical temperature). The 1997/1998 event led to remarkable impacts in the Indian Ocean: warm anomalies exceeded 2C over the West Indian Ocean and enhanced convection over Africa was anomalously strong even for EN (Webster et al., 1999; Su et al., 2001). Sea surface temperatures north of the equator were anomalously warm in the spring and summer of 1998 as well (Yu and Rienecker, 2000).

As discussed in Garfinkel et al. (2013a), the cold point moves toward India in spring and thus warming in this area can impact water vapor. We suggest that this region may have played a role in the unique evolution of this event, though further research is needed in order to to better assess the impact of Indian Ocean SST anomalies on the tropical lower stratosphere. Finally, recent work suggests that mid-tropospheric warming can directly lead to a warmer cold point tropopause and wetter stratosphere (Dessler et al., 2013, 2014). Hence, an ENSO event that more efficiently warms the mid-troposphere (such as the 1997/1998

event) can more efficiently moisten the stratosphere.

## 6   Implications for the post-2000 drop and the 2015/2016 EN event

It has been suggested that SST changes in the Indo-Pacific contributed to some of the post-2000 drop in water vapor (Rosenlof and Reid, 2008; Garfinkel et al., 2013b) via ENSO (Brinkop et al., 2016), and here we consider whether the AGCM simulations simulate



a drop. Before proceeding, it is important to mention that the 1997/1998 El Niño was followed by nearly three consecutive years of strong La Niña conditions: the Nino3.4 index in the ERSST4 dataset did not drop below -0.5K until March 2001. As discussed above, strong La Niña events also lead to moistening of the stratosphere. The net effect is that ENSO was in a phase that leads to enhanced water vapor during 1998, 1999, and 2000.

Figure 11 shows the evolution of anomalous annual averaged entry water vapor in the AGCM simulations (excluding the simulations that represent the eruption of Mt. Pinatubo). Figure 11a shows that water vapor increases with time, though with a pronounced decrease in the early 2000s as observed. Specifically the SST forcing appears to account for a drop in water vapor in the years 2002 through 2004 relative to 1998 through 2000 despite the gradual moistening trend. If we define the drop as the difference in water vapor between 2002 through 2004 and 1998 through 2000, the mean magnitude of this dehydration among

all integrations is 0.14ppmv, and the maximum drop in any of the integrations is 0.32ppmv. The mean value is approximately 23% of the total drop (which equals 0.62ppmv in the deep tropics if we apply the same definition to SWOOSH data). The evolution of cold point temperatures (defined in section 3) is overlaid on the water vapor evolution, and indicates that this drop in water vapor was due to a cooling of cold point temperatures. Hence, SST changes contributed to the drop (in agreement with Rosenlof and Reid, 2008), but were not the major forcing factor, consistent with Garfinkel et al. (2013b) and Brinkop et al.

(2016). The rest of the drop is associated with the QBO and with BDC variability (Randel et al., 2006; Fueglistaler, 2012; Fueglistaler et al., 2014; Dessler et al., 2014). The magnitude of the drop is 0.09ppmv if we consider water vapor area weighted from 60S to 60N.

    Figure 11b shows time variation of water vapor if the BDC in each integration is regressed out. The long term moistening trend is much clearer, which suggests that part of the reason water vapor concentrations have not risen over the satellite era

is cooling due to an accelerated BDC in the lowermost stratosphere (Polvani et al., 2016; Garfinkel et al., 2017). The drop in the early 2000s in Figure 11b is weaker than in Figure 11a, which suggests that part of the pathway through which SSTs led to dehydration in the early 2000s was through accelerating the BDC, or alternatively that part of the rapid acceleration of the BDC which subsequently led to less water vapor was forced by SST changes. The net effect is that ENSO variability can alias onto sub-decadal variability in water vapor.

Finally, five of the integrations have been extended to the near present and hence include the 2015/2016 El Niño event. This event was comparable in strength in the Nino3.4 region to that in 1997/1998, and while it satisfies the criteria we adopt for an EP event, it was less strongly eastern Pacific-focused as compared to the 1997/1998 event. We now consider the evolution of water vapor in those integrations in Figure 12. Figure 12a shows entry water vapor between 5S-5N, and Figure 12b shows tropical cloud ice between 5S-5N at 100hPa. The thick black line denotes the ensemble mean of the five integrations. Note that

these simulations are forced with time varying SSTs and sea ice only.

    The model simulates a 0.5ppmv increase in H2O in 2016 (annual average) as compared to 2015, approximately 70% of the observed increase. Note that part of the observed increase was due to the phase of the QBO, and the QBO phase in GEOSCCM does not match that in nature. Hence, the model is clearly capable of capturing the enhanced stratospheric water vapor following strong EN events. The seasonal evolution of the change is shown in Figure 4c, and the increase in water vapor occurs in the

spring after the EN event has already begun to decay. The moistening in 2016 is comparable to that in 1998 (cf. figure 4ac).



There is also a jump in tropical cloud ice at 100hPa of around 0.5ppmv associated with this event, in agreement with Calipso data (Avery et al., 2017), and even at 85hPa cloud ice increases by 0.05ppmv. The spatial distribution of the change in cloud ice at 85hPa in December 2015 is shown in Figure 12c; the pattern of anomalous ice matches that found in Calipso data (see Figure 1 of Avery et al., 2017). Note that these integrations also simulate a drop after 2011 (Urban et al., 2014; Gilford et al., 2016), suggesting that part of this drop was forced by SSTs as well. Hence, in summary, strong EN events regardless of their type lead to an enhanced water vapor response in both GEOSCCM and in nature.

## 7 Conclusions

Tropical lower stratospheric temperature and water vapor changes have important implications for both stratospheric and tropospheric climate as well as stratospheric ozone chemistry (SPARC-CCMVal, 2010; World Meteorological Organization, 2011, 2014). Hence, it is crucial to understand interannual changes in this region in order to correctly interpret future changes. Analysis of a series of chemistry-climate atmospheric model in two distinct configurations - coupled to an interactive ocean model and forced by historical sea surface temperatures - yielded the following conclusions:

1. The impact of El Niño-Southern Oscillation in this region is nonlinear in spring. While moderate El Niño events lead to cooling in this region, strong El Niño events appear to lead to warming, even as the response of the large scale Brewer Dobson Circulation appears to scale nearly linearly with El Niño. The tropospheric warming associated with strong El Niño events extends into the tropical tropopause layer and up to the cold point, where it allows for more water vapor to enter the stratosphere. The net effect is that both strong La Niña and strong El Niño events lead to enhanced entry water vapor and stratospheric moistening in spring. Only in midwinter is the response linear.

2. There is no appreciable difference in the tropical lower stratospheric response to Central Pacific versus Eastern Pacific El Niño events, if one controls for the amplitude of the El Niño event.

3. The very strong El Niño event in 1997/1998 followed by more than two consecutive years of La Niña led to enhanced lower stratospheric water vapor. As this period ended in early 2001, entry water vapor concentrations declined. The magnitude of this effect is 0.14ppmv, which accounts for approximately 23% of the observed water vapor drop.

In light of these results, we wish to emphasize that multiple linear regression approaches to attributing stratospheric variability in water vapor and temperature to forcings such as ENSO are problematic, as the stratospheric response to ENSO is nonlinear in the tropical lower stratosphere. Second, we wish to caution that compositing approaches of ENSO into Central Pacific and Eastern Pacific types can sometimes lead one astray, as Central Pacific El Niños are weaker. Specifically, Garfinkel et al. (2013a) compared EP EN events including 1997/1998 to all CP EN events. A more meaningful comparison is EP EN events excluding 1997/1998 to all CP EN events, and our GEOSCCM experiments suggest that there is no difference in stratospheric response for such a comparison of composites. Finally, it is important to consider SST variability when considering decadal variability in the lower stratosphere.



This study leaves several unanswered questions. First, it is not clear mechanistically how upper tropospheric warming leads to moistening of the stratosphere. Second, entry water vapor may be influenced by physical processes that are missing or poorly-represented by the current generation of climate models, and hence similar studies must be performed as models improve. However, the nonlinearity of the lower stratospheric response to El Niño is robust in both observations and in the model, and hence caution must be exercised when deciding on a methodology for analyzing the tropical stratospheric response to El Niño.

## 8    Appendix: Linearity of the response of the BDC to ENSO

The text argues that the nonlinearity in the lower stratospheric response to ENSO arises not from the large scale stratospheric BDC, but rather from processes below in the tropical tropopause layer. We now provide additional support for the linearity of the large scale stratospheric BDC to ENSO. Figure 13 and 14 shows the response in tropical mean age at 50hPa and 70hPa and NH polar cap temperature at 85hPa to ENSO in the coupled ocean-atmosphere and AGCM experiments respectively. The response of mean age is linear with respect to ENSO in all seasons at these levels. The NH polar stratospheric response to ENSO is also linear in these experiments, and displays little sensitivity to the flavor of the EN event: both CP and EP EN events lead to polar stratospheric warming (in agreement with Garfinkel et al., 2012). There is little difference in the response between the coupled ocean-atmosphere simulations and the AGCM simulations.

These changes in the BDC are ultimately driven by upward propagating waves from the troposphere driven by ENSO, and the changes in these waves are diagnosed by examining the 100hPa heat flux response to ENSO (see Figure 15 for the AGCM simulations; the daily data to compute the heat flux for the coupled integrations is not available). We average the heat flux from 25° to 75° in each hemisphere following Fueglistaler et al. (2014), and the SH is shown in the left column and the NH is shown in the middle column. In both hemispheres heat flux increases during EN (in the SH heat flux is negative climatologically), and the changes in heat flux are also linear with respect to ENSO. The right column of Figure 15 shows the wavenumber 1 heat flux from 40° to 80°, the wave driving of most importance for the polar warming in response to EN (Garfinkel and Hartmann, 2008). It too changes linearly in response to ENSO.

The net effect is that EN leads to more wave driving of the BDC, leading to an accelerated BDC and cooling of the tropical lower stratosphere via enhanced adiabatic cooling. This adiabatic cooling must be balanced by anomalous diabatic heating, as the thermodynamic budget must balance. The way in which this balance occurs is through anomalous longwave heating. To confirm this, Figure 16 compares the diabatic heating due to longwave and shortwave radiation. The temperature and diabatic heating for the coupled ocean-atmosphere integration is shown in Figure 1. The colder temperatures present during EN emit less radiation, and thus the usual longwave cooling is now weaker than before. This manifests as a positive longwave heating anomaly. Quantitatively, the slope in total diabatic heating and from longwave diabatic heating is nearly identical, though the slope for longwave is slightly larger. It does not appear to matter which flavor of ENSO event is occurring (red, black, and blue markers all line up). The slope is largest in JF, likely because the subpolar NH heat flux is affected most strongly in JF (right column of Figure 15). EN causes the shallow branch to accelerate in all seasons (Simpson et al., 2011).





The second column of figure 16 shows shortwave diabatic heating. For shortwave, EN leads to a slight cooling of the atmosphere. This could be because EN leads to less ozone (by $\sim 5\%$ per K due to an accelerated BDC as found by Oman et al 2013), and hence there is less shortwave heating. However, this effect is far smaller than the longwave effect, so it is of secondary importance. However, it could explain the slight difference in slope between DTDT and DTDTLW , as the slopes

5   for DTDTLW and DTDTSW (left and middle of figure 16) add up to that for DTDT (middle of Figure 1).



Table 1: GEOSCCM Model Experiments

| ocean | forcings | integration length | reference |
|---|---|---|---|
| coupled ocean | 1950 timeslice | 340 (240) | Li et al. (2016) |
| historical SSTs | SST+sea ice | 13x(1980–2009) | 5 from Garfinkel et al. (2015), Aquila et al. (2016) + 5 new |
| historical SSTs | SST+sea ice GHG | 3x(1980–2009) | Aquila et al. (2016) |
| historical SSTs | SST+sea ice+GHG+ODS | 19x(1980–2009) | Aquila et al. (2016), Garfinkel et al. (2015)+6 new |
| historical SSTs | SST+sea ice+GHG+ODS+volcanoes | 3x(1980–2009) | Aquila et al. (2016) |
| historical SSTs | SST+sea ice+GHG+ODS+volcanoes+solar | 4x(1980–2009) | Aquila et al. (2016) + CCMI |



Table 2: Events composited for AGCM and observations

| composite | years |
|---|---|
| EP El Niño | 1982/1983, 1986/1987, 1991/1992, 1997/1998, 2015/2016 |
| CP El Niño | 1994/1995 and 2004/2005 |
| EP La Niña | 1984/1985, 1995/1996, 1999/2000, 2005/2006, 2007/2008 |
| CP La Niña | 1983/1984, 1998/1999, 2000/2001, 2008/2009 |



**Figure 1.** Seasonally resolved anomalies in the tropical lower stratosphere stratified by the Nino3.4 index in NDJF in the coupled ocean-atmosphere GEOSCCM integration. (a-c) November and December; (d-f) January and February; (g-i) March and April; (j-l) May and June. (left) temperature at 85hPa, 5S-5N; (center) diabatic heating rate at 70hPa, 5S-5N; (right) water vapor at 85hPa, 5S-5N. For all quantities, the data has been detrended (see section 3) and the component of the variance linearly associated with the QBO at 50hPa two months prior has been removed. Winters categorized as Central Pacific ENSO are in black, Eastern Pacific ENSO are in cyan, and all other years in red. A linear least-squares best fit is shown in each panel, and the slope is indicated.





**Figure 2.** As in Figure 1 but for 42 AGCM GEOSCCM integrations. The ensemble mean response is indicated with a large x, and each integration with a dot.





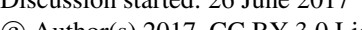

**Figure 3.** As in Figure 1 and 2 but for the MERRA reanalysis (left and center columns) and SWOOSH (right column).



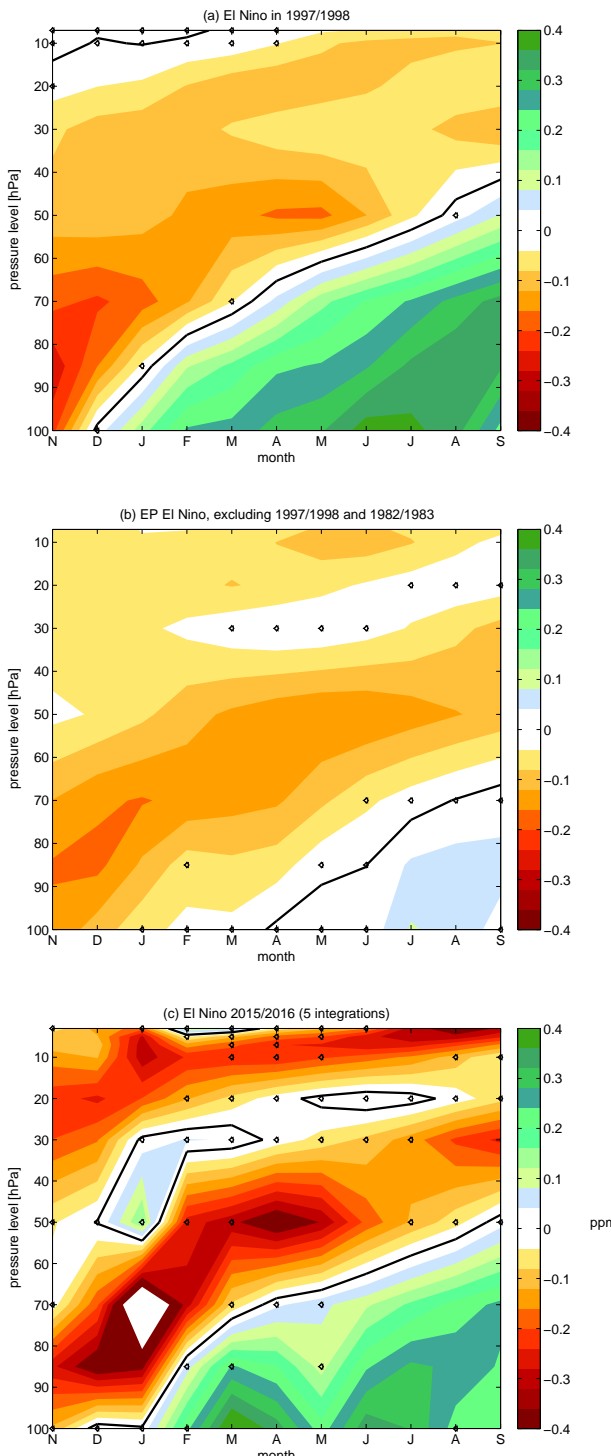

**Figure 4.** Water vapor anomalies (ppmv) for the 42 GEOSCCM integrations in (a) 97/98; (b) all EPEN events except 97/98 and 82/83; (c) 2015/2016 in the 5 experiments that have been extended to the near present. Anomalies that are not significant at the 95% level are marked by black symbols. The effect of the QBO at 50hPa and the linear trend have been linearly regressed out of all anomalies.





**Figure 5.** Temperature anomalies (Kelvin) at 100hPa for 1997/1998 in the GEOSCCM AGCM integrations in November/December (top) through the following May/June (bottom). Anomalies that are not significant at the 95% level are marked by black symbols. The green and magenta contours denote specific cold isotherms in the climatology and for this specific composite in order to highlight the location of the cold point. The effect of the QBO at 50hPa and the linear trend have been linearly regressed out of all anomalies. The contour interval is 1/3K.





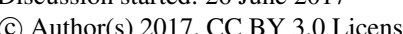

**Figure 6.** As in figure 5 but for all ENSO events except 1982/1983 and 1997/1998 in the GEOSCCM AGCM integrations.



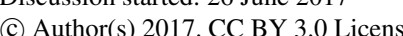

**Figure 7.** Tropical (5S-5N) temperature anomalies (Kelvin) for (a-d)for all EP EN events excluding 97/98 and 82/83, (e-h) 97/98, and (i-l) 82/83, in the AGCM simulations. Anomalies that are not significant at the 95% level are marked by black symbols. The green and magenta contours denote specific cold isotherms in the climatology and for this specific composite in order to highlight the location of the cold point. The effect of the QBO at 50hPa and the linear trend have been linearly regressed out of all anomalies. The contour interval is 0.3K.







**Figure 8.** As in figure 7 but for the MERRA. Note that the 1982/1983 EN was preceded by the eruption of El Chicon, and thus the stratosphere is anomalously warm in the right column.



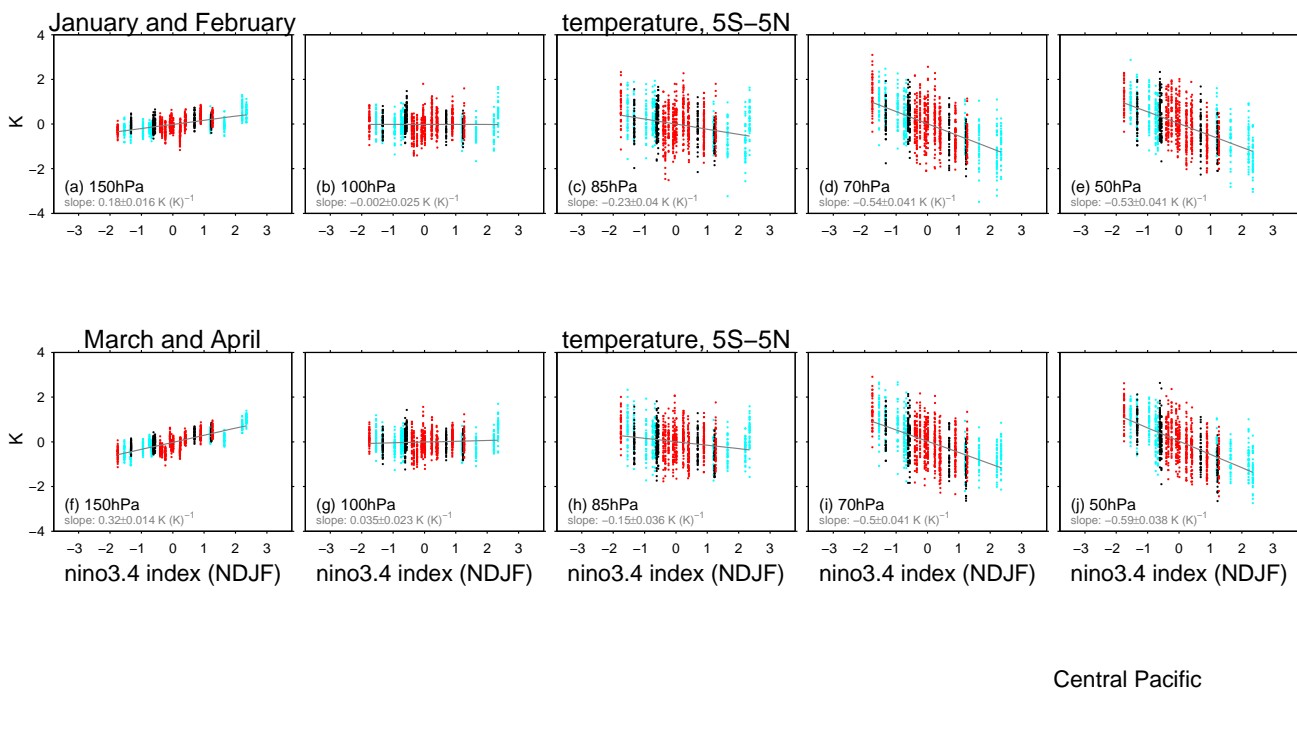

**Figure 9.** tropical (5S-5N) temperature detrended anomalies in the AGCM simulations for the tropical tropopause layer and lower strato-sphere stratified by the value of the Nino3.4 index in NDJF for (top) Winter (January/February) and (bottom) early spring (March/April).





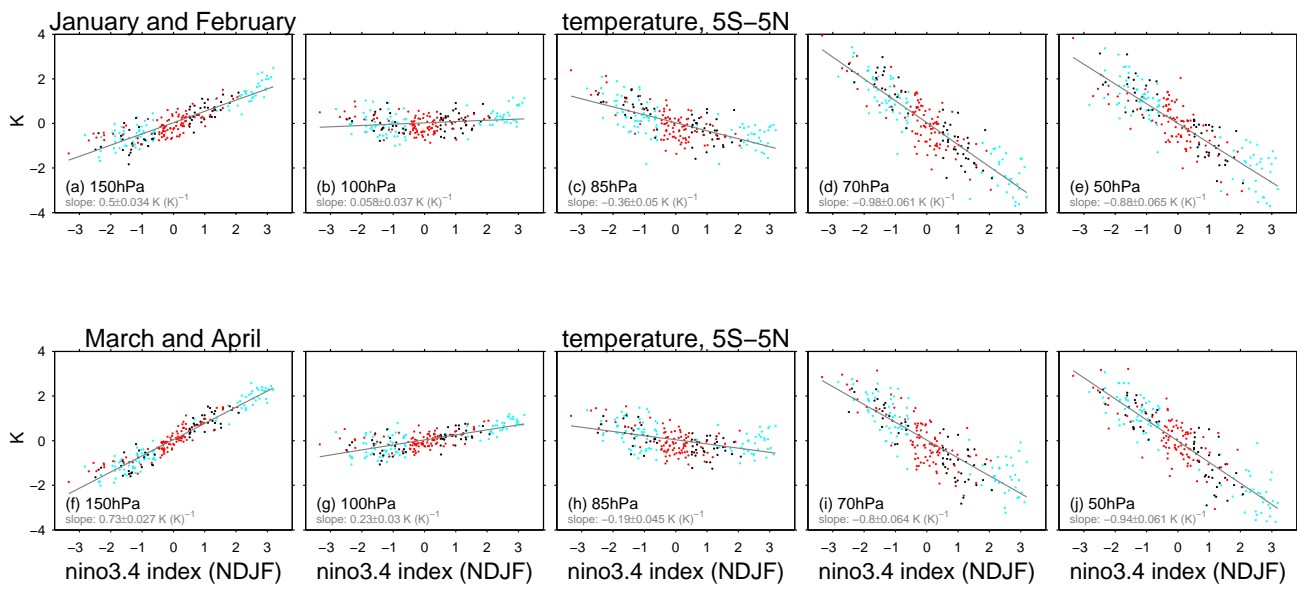

**Figure 10.** As in Figure 9 but for the coupled ocean-atmosphere simulation.







**Figure 11.** Annual average water vapor anomalies at 85hPa (cyan) and cold point temperature anomaly (black) for the GEOSCCM AGCM experiments (a) without regressing out the BDC and (b) with regressing out the BDC. No detrending has been performed on any of the timeseries. The cold point temperature is defined in the methods section.





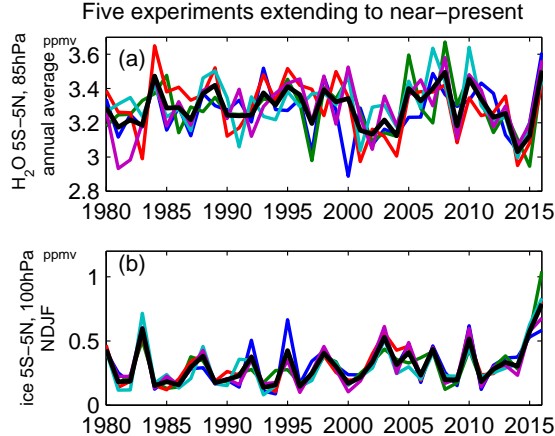

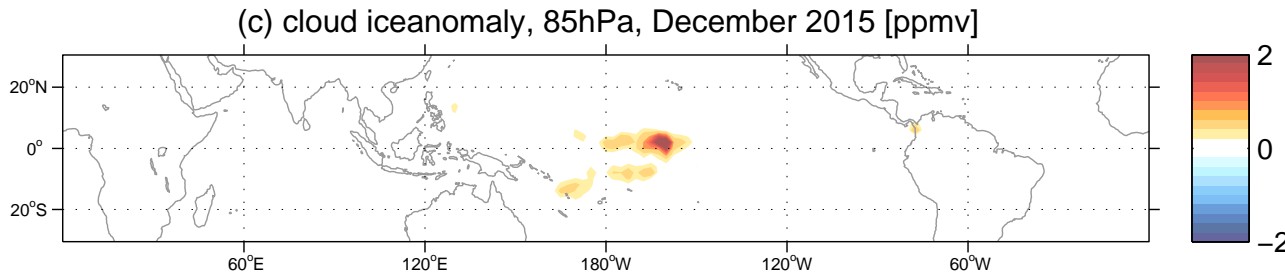

**Figure 12.** Annual average water anomalies at 85hPa for the 5 GEOSCCM AGCM experiments which can be extended to the near-present (a) 85hPa water vapor between 5S and 5N. (b) Cloud ice at 100hPa between 5S and 5N. A black line indicates the ensemble mean, and color indicates each of the simulations. No detrending has been performed on any of the timeseries. (c) Cloud ice anomalies at 85hPa in December 2015 in the ensemble mean.







**Figure 13.** As in Figure 1 but for anomalies in mean age at 50hPa and 70hPa (left and center) and NH polar cap temperature changes at 85hPa (right).





**Figure 14.** As in Figure 13 but for 42 AGCM GEOSCCM integrations. The ensemble mean for each EN event is indicated with a large x.





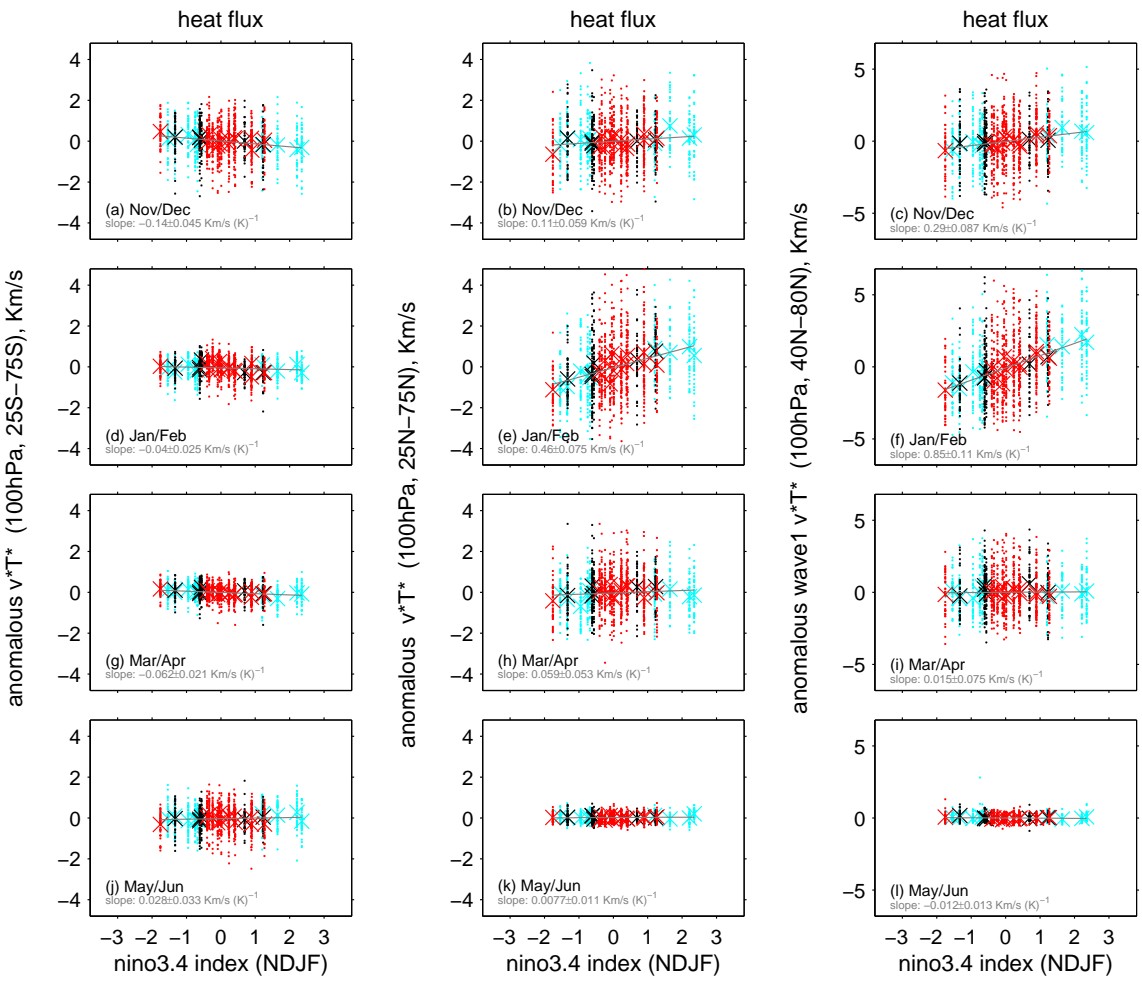

**Figure 15.** Heat flux anomalies for ENSO events in the AGCM GEOSCCM run. The left column is for SH heat flux area weighted from 25S to 75S (note that negative values indicate more heat flux in the SH), the middle column is for NH heat flux area weighted from 25N to 75N, and the right column is for NH zonal wavenumber 1 heat flux area weighted from 40N to 80N.




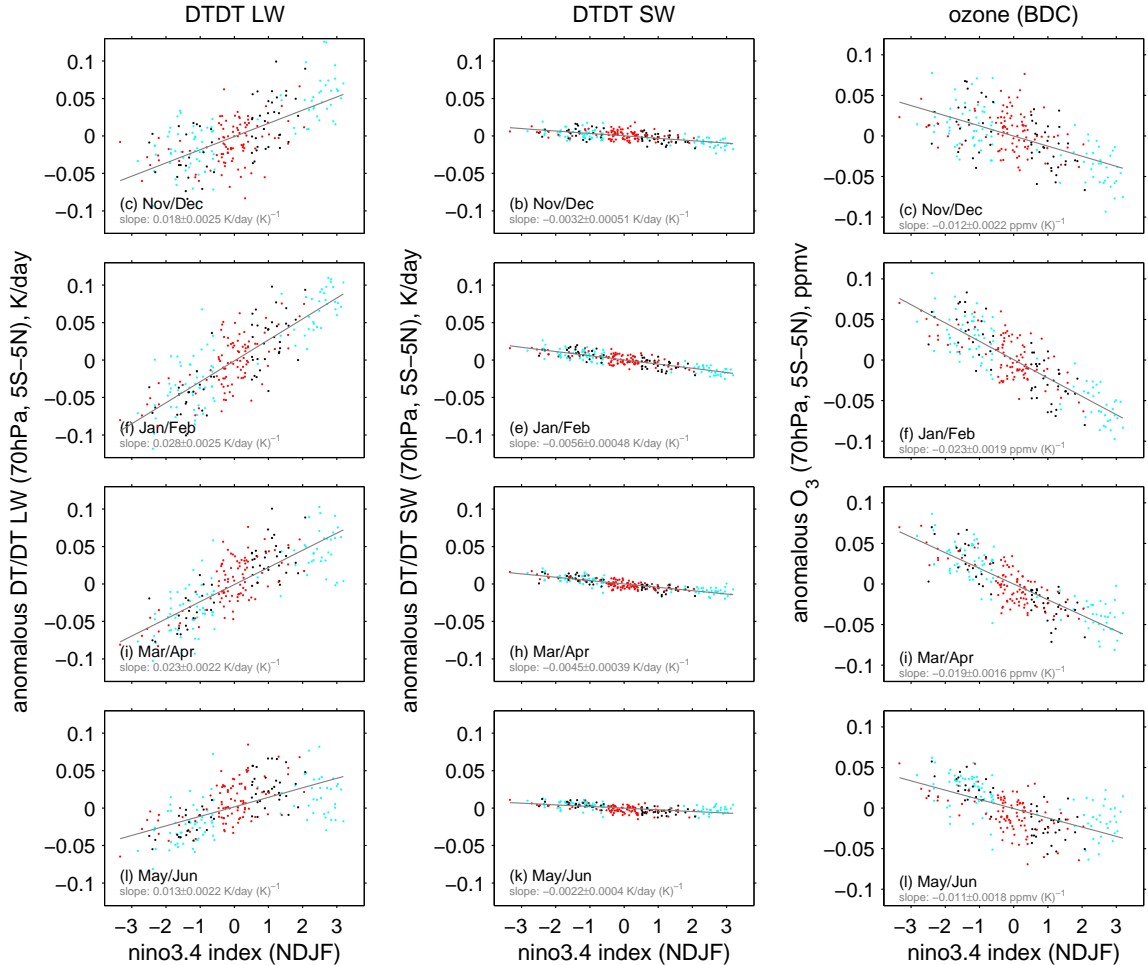

**Figure 16.** BDC anomalies for ENSO events in the last 240 years of a coupled ocean-atmosphere run.





*Acknowledgements.* CIG was supported by the Israel Science Foundation (grant number 1558/14) and by a European Research Council starting grant under the European Union's Horizon 2020 research and innovation programme (grant agreement No 677756). We thank those involved in model development at GSFC-GMAO, and support by the NASA MAP program. We thank Valentina Aquila for performing some of the experiments discussed here, and Darryn W Waugh and Margaret M Hurwitz for suggestions. High-performance computing

5 resources were provided by the NASA Center for Climate Simulation (NCCS). Correspondence and requests for data should be addressed to C.I.G. (email: chaim.garfinkel@mail.huji.ac.il). El Niño indices based on the ERSSTv4 data were downloaded from cpc.ncep.noaa.gov/data/indices/ersst4.nino.mth.81-10.ascii.



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
