# Peer review of "Nonlinear response of tropical lower stratospheric temperature and water vapor to ENSO"

_Atmospheric Chemistry and Physics, 2017_

## Referee Comment (RC1) · Anonymous Referee #1 · 29 Aug 2017

General comments: This paper discusses the impact of ENSO on the tropical lower-stratospheric (LS) temperature and water vapor by analyzing datasets composed of numerical simulations and reanalyses. The authors found that both La Nina (LN) and strong El Nino (EN) lead to wet stratosphere while moderate EN leads to dry stratosphere even though the strength of stratospheric Brewer-Dobson circulation responds linearly to EN. The nonlinearity, i.e., the increase of ST water under strong EN condition is interpreted as the tropospheric warming extending up to the cold point that regulates the water entry to the stratosphere. The strong EN in 1997/98 and the following LN are attributed to the cause of the drop of ST water vapor in early 2001.

1. The analyses are limited to the temperature response and there found no argument on the modulation of pathways for the air entering the stratosphere. The coldness of

the tropopause region does not necessarily result in the stratospheric dryness; as was pointed out by Bonazzola and Haynes (2004), "the sampling effect" as well as "the temperature effect" must be considered.

2. As for the millennial ST water drop, Fueglistaler (2012) and Hasebe and Noguchi (2016, ACP) identified its occurrence as October 2000 and September 2000, respectively. The current authors' mentioning of the year 2001 is different from these studies. Some arguments are required on the difference in the occurrence time and, most importantly, the driving mechanism.

3. It is not clear how "anomalies" and means are defined in many variables such as LS temperature, SST, heating rate etc. "Anomalous" labeled for vertical axes is not appropriate.

4. The authors' notion of nonlinearity is evident only in those shown in cyan with the Nino3.4 index greater than 2 (Figs. 1, 2, 9, and 10). Those points having the index > 2 will correspond to 1997/98 EP EN (p.6, l.30). In this context, it is important to study the features for this specific event in Section 5. The suggestion on the impact in the Indian Ocean is interesting, but there is no conclusive evidence having been shown.

5. The argument in Section 6 is not convincing. The time series of H2O and cold point temperature show large negative anomalies in 1997 followed by large positive anomalies in 1998. The H2O drop is more pronounced (anomalies are larger in magnitude of negative values) in 1997 than in 2001, but there found no discussion on the cause of large drop in 1997. I don't find any logical consequence in the statements given in page 9, lines 13 to 16.

6. Appendix: I don't understand why the mean age is discussed in the context of this paper. In addition, there found no explanation on how the mean age is estimated.

The bottom line will be that the LS water vapor tends to decrease in response to El Nino quantified by Nino 3.4 index but that the strong 1997/98 El Nino was exceptional

in that it caused LS water increase. It is not clear if it is due to the warming in the TTL or it is related to the "flavor" (or type) of EP category it was classified. The mechanism has not been made clear by this analysis; it remains in the level of speculation. The terminology of "nonlinear response" may not be wrong, but it does not help understand the nature of LS water response to EN. The authors have interesting dataset obtained from ensemble runs, but it has not been analyzed satisfactorily. I recommend total rewrite of the manuscript after conducting analyses focusing on the specific features on 1997/98 EP EN. Considering the time necessary for the analysis, I suggest withdrawal of the present manuscript to consider re-submission.

Specific comments:

p.6, l.14-15: "EN leads to strong cooling" will be OK, but "LN leads to warming" is not obvious since the vertical axis is anomalies.

p.6, l.29: "This is especially evident in Figure 1il": In Figure 1i, the negative value of slope appears statistically significant, which may be interpreted as a linear response.

Figure 4. The choice of green (+) and red (-) in color scale is confusing; the choice of the same color as in other figures (Figs. 5, 6 and so on) is recommended.

Figures 5 and 6: The distribution of cold region is only one aspect of TTL dehydration. There is no information on the location of the dehydration that is taking place for the air entering the stratosphere. The distribution of Lagrangian cold point was reported to have changed dramatically during 1997/98 El Nino (Fig. 8 of Hasebe and Noguchi, 2016).

p.10, l.5-6: "regardless of their type": It seems the nonlinearity appears only in EP type El Nino (2015/16 is also categorized as EP EN).

Technical corrections are omitted because a whole rewrite is suggested.
* * *

---

## Referee Comment (RC2) · S. Fueglistaler (Referee) · 6 Sep 2017

Garfinkel et al. study the effect of ENSO on temperature around the tropical tropopause, and related to that on water entering the stratosphere. They study the problem using observations of stratospheric water vapor (the SWOOSH data set), temperatures from the MERRA reanalysis, and climate model simulations in a range of configurations. The paper's main point is that the response in tropopause temperature and water entering the stratosphere is non-linearly related to ENSO (as represented by some index) - such that both strong El Nino and La Nina lead to a temperature increase (and correspondingly moistening of the stratosphere). Using this result, they argue that the sequence of strong El Nino followed by strong La Ninas in the late 1990's led to elevated temperatures (and moister air) for a few years, which contributes to the

'drop' observed around September/October 2000. The hypothesis put forward is very interesting - but I have a number of concerns.

I could not help the impression that this paper was written up slightly careless. At times, the text reads more like a story than a scientific paper; similarly, the paper has problems finding the right tone. Consider the abstract. There, it is first written that: "The impact ... is nonlinear." which leaves no room for doubt. However, the next sentence does not provide the hard evidence expected by the reader, but uses the rather weak word "appear" twice. Also, considering the seemingly straightforward hypothesis - the non-linearity of ENSO - I expected to be shown a plot that shows the non-linearity beyond doubt. Instead, the paper presents a full 16 figures that show a lot of information - most of it only qualitatively discussed. The paper would be much stronger if the authors were able to support the main point of their paper in one or two clearly drafted figures. Figure 1 presumably presents the model data that best supports the argument for non-lineariy - however the points are so small, and cyan has very little contrast, such that it is easy to overlook the datapoints supporting the hypothesis. Simply tweaking colors and symbol size would probably help. Also, I'd like to see a more quantitative treatment of the non-linearity (i.e. it would be simply to compare the linear regression with a non-linear regression). Also, it would be fair to show the statistical uncertainty in the "observational" data shown in Figure 3; we should be honest that the observational timeseries is really (too?) short to make statistically robust statements.

The paper then applies the argument of non-linearity to explain the sudden and persistent drop of stratospheric water vapor around October 2000. However, there is a major conundrum pointed out in Fueglistaler et al. (J. Geophys Res., 2013) that needs to be addressed here: The arguably best representation of true temperatures in re-analysis data fails to properly produce a drop as observed in HALOE data. That is, the mechanism discussed in this paper applies to the large-scale effect of temperature and circulation, but the problem is that even if the free-running GCM would recover the reanalysis temperature perfectly, it would not be able to produce a drop as prominent

as observed by HALOE. Correspondingly, it is not surprising that the drop diagnosed by the authors is only 23% of the HALOE drop. While there is plenty of good reason to have trust in HALOE data, it is crucial to note here that the stratospheric water vapor time series as observed by SAGEII agrees very well with the reanalysis-based model estimates (Fueglistaler et al., 2013). This needs to be discussed; and I would strongly encourage you to also consider quantifying the importance not against HALOE, but against the AMIP-mode model generated data (this helps your paper). However, the analysis of the drop as presented in Figure 11 is close to cherry-picking: anyone can see that what is labelled here as "decadal drop" is anything but a decadal drop. I also suspect that this time series does not compare favourably against SWOOSH at all - should this not be reason for serious concern given that this is an AMIP run?

To summarize, the paper by Garfinkel et al. touches many interesting points, and makes use of interesting numerical model runs. The paper needs, however, a major overhaul; the main points need to be worked out clearer in the data, and the discussion of the "drop" needs to be more careful. Given the recommendation for major revisions, I do not go further into the details of the current manuscript.

Signed review, S. Fueglistaler

---

## Author Comment (AC1) · 19 Oct 2017

Dear Editor

Thank you for editing our manuscript. In response to the reviewers' comments, we have re-written large segments of the manuscript. Specifically,

1. Much more attention is devoted to demonstrating the key role of Indian Ocean SSTs for the springtime tropical lower stratospheric response to El Nino.
2. Nonlinearity is now quantified.
3. Eight figures (out of the original sixteen) and their associated discussion have been removed in order to minimize distraction from the key points. Five of the eight now appear in supplemental material, and three have been eliminated entirely.
4. The discussion of the millennium drop has been shortened and focused.

The newly added text on the importance of the Indian Ocean is quite long, and for clarity it is copied below rather than within the detailed responses to the reviewers. The revised manuscript is ready for uploading, and we await your instructions.

Sincerely
Chaim Garfinkel (on behalf of the coauthors)

Why was the 97/98 El Niño tropospheric warming so distinct from other events? While this was the strongest El Niño over the period considered by this paper, the 1982/1983 El Niño was not much weaker than the 1997/1998 event as measured by the Nino3.4 index, yet the impact of the 1982/1983 on water vapor was qualitatively different. Furthermore, the upper tropospheric warming in the Central and East Pacific sectors for the 1982/1983 and 1997/1998 events (Figure 8) are similar. This suggests that the Central and East Pacific responses cannot explain the difference in stratospheric response. In contrast, these two events differed quite dramatically in the Indian Ocean (and more generally in zonally averaged tropical temperature). The 1997/1998 event led to remarkable impacts in the Indian Ocean: warm anomalies exceeded 2C locally over the West Indian Ocean and enhanced convection over Africa was anomalously strong even for EN (Webster et al., 1999; Su et al., 2001). Sea surface temperatures north of the equator were anomalously warm in the spring and summer of 1998 as well (Yu and Rienecker, 2000). As discussed in Garfinkel et al. (2013a), the cold point moves toward India in spring and thus warming in this area can impact water vapor. This difference in near surface conditions in the Indo-Pacific and Nino3.4 region is quantified in Figure 1. Conditions during the 1982/1983 event are shown with a red diamond, and during the 1997/1998 event with a large red x. Despite largely similar anomalies in the Nino3.4 region, the 1997/1998 event was characterized by remarkably warm anomalies in the Indo-Pacific that lie in the tail of the warming generated spontaneously in the coupled ocean-atmosphere model.

The importance of Indian Ocean SSTs for entry water vapor is quantified in Figure 9, which shows the regression coefficient between 85hPa water vapor and 2meter temperatures from 5S to 5N at each longitude grid point. We show both the regression coefficient in the annual average with no lag between water vapor and surface temperature and in springtime with 2meter temperatures leading water vapor by two months (Garfinkel et al., 2013a). The black curve shows the regression after linearly regressing out the BDC from the water vapor, and the blue curve regression after linearly regressing out the QBO from the water vapor.

In the annual average, warmer near-surface temperatures over the Central and Eastern Pacific lead to dehydration of the stratosphere in all three data sources, though during spring warming in the eastern Pacific leads to moistening of the stratosphere two months later. More importantly however, stratospheric water vapor is most sensitive to variability in the Indian Ocean basins and the Warm Pool region, with warmer temperatures in this region leading to enhanced water vapor in all three data sources in spring (and if the BDC influence on water vapor is regressed out, also in the annual average). Results are similar if correlations are examined (not shown).

Figure 10 demonstrates that the nonlinearity of the spring stratospheric response to EN is due to Indo-Pacific surface temperatures. It is constructed similarly to Figure 2, but years are stratified by 2meter temperatures from 50E to 150E instead of by the Nino3.4 index. Instead of the pronounced springtime nonlinearity evident in Figure 2, the lower stratospheric response to Indo-Pacific surface temperature is linear in all seasons. In winter, a warmer near surface leads to impacts similar to that of ENSO (compare top row of Figure 2 to 10). In March and April, on the other hand, a warmer Indo-Pacific leads to an accelerated BDC and a colder lower stratosphere, but to no robust changes in water vapor. In May and June, a warmer Indo-Pacific still leads to an accelerated BDC, but despite this accelerated BDC the lower stratosphere moistens. Results are similar for

the AMIP integrations (not shown), with the 1997/1998 event leading to lower stratosphere moistening despite an accelerated BDC.

In summary, an ENSO event that more efficiently warms the mid-troposphere (such as the 1997/1998 event) by modifying SSTs in the Indian Ocean as well can more efficiently moisten the stratosphere. Strong EN events tend to have a stronger impact on the Indian Ocean than more moderate events (cf. Figure 1), and this tendency accounts for the nonlinearity in the impact of EN on the spring tropical lower stratosphere.

[Figure]

**Figure 1.** Relationship between near surface temperatures over the Nino3.4 region and over the Indian Ocean and Warm pool region from 5S to 5N in (blue) the GEOSCCM coupled ocean-atmosphere integration and in (red) MERRA reanalysis data. The EN event 1982/1983 is indicated with a large red diamond, and the EN event in 1997/1998 is indicated with a large x. (left) January through April; (right) annual average.

[Figure]

**Figure 9.** Regression coefficent between tropical (5S-5N) T2m and zonally averaged entry water vapor at 85hPa in (a-b) the last 240 years of a coupled ocean-atmosphere run; (c-d) the AMIP runs; (e-f) for SWOOSH water varpor and MERRA 2 meter temperatures. The longitude bands corresponding to the Indian Ocean, Nino3, and Nino4 regions are in color. The left column is for annual averaged quantities and the right column is for springtime (March through June) water vapor with T2m two months prior.

[Figure]

**Figure 10.** As in Figure 2 but stratifying years based on 2meter temperature from 50E to 150E, 5S-5N.

Reviewer #1

General comments: This paper discusses the impact of ENSO on the tropical lower-stratospheric (LS) temperature and water vapor by analyzing datasets composed of numerical simulations and reanalyses. The authors found that both La Nina (LN) and strong El Nino (EN) lead to wet stratosphere while moderate EN leads to dry stratosphere even though the strength of stratospheric Brewer-Dobson circulation responds linearly to EN. The nonlinearity, i.e., the increase of ST water under strong EN condition is interpreted as the tropospheric warming extending up to the cold point that regulates the water entry to the stratosphere. The strong EN in 1997/98 and the following LN are attributed to the cause of the drop of ST water vapor in early 2001.

1. The analyses are limited to the temperature response and there found no argument on the modulation of pathways for the air entering the stratosphere. The coldness of the tropopause region does not necessarily result in the stratospheric dryness; as was pointed out by Bonazzola and Haynes (2004), "the sampling effect" as well as "the temperature effect" must be considered.

Thank you for pointing out this important effect. We agree that both sampling effects and temperature effects are important for TTL dehydration, and GEOSCCM of course includes both

effects. However, we only have once-daily (daily averaged) output from the model on limited pressure levels and not on full model levels, and hence we are unable to quantify the sampling effect by running trajectories. We are therefore limited to analyzing temperature effects in our attempt to explain mechanistically why water vapor changes in the way it does in the GEOSCCM simulations.

Perhaps more importantly, there is quite a lot of scatter in our figures about the forced response, and this scatter represents (in part) the sampling effects to the final water vapor concentrations in the stratosphere. Stated another way, El Nino directly forces large scale changes in wind patterns and temperatures, and these effects are captured in the forced response as identified by the mean of the 42 ensemble members; the wind and temperature patterns in any specific integration among the 42 will differ from all of the others, and these deviations in the wind/temperature pattern are what we try to filter out by forming a large ensemble.

One could imagine that the sampling effects identified by Bonazzola and Haynes 2004 are due to such unforced variability that happened to be present in 1998 and 1999, and were not actually forced by the underlying ENSO event. However this hypothesis needs to be tested, and we don't mention this possibility in the text.

We have added the following to the data section: "The output necessary to run a trajectory model was not archived, and hence we cannot quantify the specific location of dehydration."

We have added the following to the conclusions in the list of unanswered questions: "Model output necessary to run a trajectory model was not archived, and hence we cannot directly address whether EN modifies the residence time in the coldest region of the tropical tropopause layer, an effect found to be important by Bonazzola and Haynes 2004. However, these sampling effects are included implicitly in GEOSCCM, and some of the diversity in response among the 42 ensemble members to an identical SST forcing is almost certaintly due to such sampling effects."

2. As for the millennial ST water drop, Fueglistaler (2012) and Hasebe and Noguchi (2016, ACP) identified its occurrence as October 2000 and September 2000, respectively. The current authors' mentioning of the year 2001 is different from these studies. Some arguments are required on the difference in the occurrence time and, most importantly, the driving mechanism.

Figures 14 and 15 in the initial submission were based on annual averaged values of water vapor. It is evident from figure 15 that individual integrations disagree about the specific year of the drop, and hence we don't expect GEOSCCM to capture the monthly variability in water vapor that was observed. Note that the QBO phase differs in GEOSCCM as compared to that observed, and the individual wave events driving the BDC differ as well.

We now clarify that these figures are based on annual timescales, and that the drop is fully consistent with the timing of the observed drop in these two publications. We also include a preamble that clarifies that the phase of ENSO from 1998 through 2004 was appropriate for a drop in late 2000.

3. It is not clear how "anomalies" and means are defined in many variables such as LS temperature, SST, heating rate etc. "Anomalous" labeled for vertical axes is not appropriate.

We now define anomalies in the methods section as the deviation from the monthly climatology for each data set.

We assume the reviewer is referring to "anomalous" on the vertical axis for figures 1,2, 3, 13-16, and this word has been removed.

4. The authors' notion of nonlinearity is evident only in those shown in cyan with the Nino3.4 index greater than 2 (Figs. 1, 2, 9, and 10). Those points having the index > 2 will correspond to 1997/98 EP EN (p.6, l.30). In this context, it is important to study the features for this specific event in Section 5. The suggestion on the impact in the Indian Ocean is interesting, but there is no conclusive evidence having been shown.

Central Pacific events tend to be weaker (which the model captures) as the reviewer points out. However the regression coefficients (which use both types) are generally not sensitive to whether we exclude or include EP vs. CP events, though indeed there are a few cases where a linear regression would suffice for CP events but not for EP events. Hence we now note that the nonlinearity is less detectable for CP events both in the results section and in the conclusions.

More generally, we now quantify much more explicitly the importance of the Indian Ocean SSTs for the nonlinearity (see above). Specifically we have added three new figures and accompanying text, while removing some of the previous figures that are less important.

5. The argument in Section 6 is not convincing. The time series of H2O and cold point temperature show large negative anomalies in 1997 followed by large positive anomalies in 1998. The H2O drop is more pronounced (anomalies are larger in magnitude of negative values) in 1997 than in 2001, but there found no discussion on the cause of large drop in 1997. I don't find any logical consequence in the statements given in page 9, lines 13 to 16.

In response to your comments and that of the second reviewer, we have rewritten the first half of section 6. At the beginning of section 6 we now clarify our expectations for the ability GEOSCCM to capture observed water vapor variability. Specifically, we note that the QBO and BDC in GEOSCCM does not match that in observations, and hence there should not be any specific expectation that the specific timing of drops should match that observed.

Rather, these experiments are useful for one purpose: quantifying the contribution of SSTs to the drop in 2000, and we limit our discussion to this point in the first half of section 6 (i.e. before we move onto 2015/2016). That the contribution is relatively minor (23%) is consistent with the

previous work showing that other forcing factors (such as the BDC and QBO) are more important for entry water vapor variability. However other studies have suggested that SSTs play a role, but the literature lacks a clear quantification of their role.

In addition to these caveats that have now been added, we have rewritten and shortened the part of section 6 that relates to the millennium drop, in order to de-emphasize this relatively short analysis as compared to the rest of the paper.

6. Appendix: I don't understand why the mean age is discussed in the context of this paper. In addition, there found no explanation on how the mean age is estimated.

Mean age is calculated by a passive tracer whose concentration increases linearly with time. We now refer to Garfinkel et al 2017 where a full paragraph is devoted to details of this calculation.

We believe it is important to establish the linearity of the large scale BDC response, and hence we include this column. As we agree this is somewhat ancillary to the main point of this paper, it has been moved to supplemental material.

The bottom line will be that the LS water vapor tends to decrease in response to El Nino quantified by Nino 3.4 index but that the strong 1997/98 El Nino was exceptional in that it caused LS water increase. It is not clear if it is due to the warming in the TTL or it is related to the "flavor" (or type) of EP category it was classified. The mechanism has not been made clear by this analysis; it remains in the level of speculation. The terminology of "nonlinear response" may not be wrong, but it does not help understand the nature of LS water response to EN. The authors have interesting dataset obtained from ensemble runs, but it has not been analyzed satisfactorily. I recommend total rewrite of the manuscript after conducting analyses focusing on the specific features on 1997/98 EP EN. Considering the time necessary for the analysis, I suggest withdrawal of the present manuscript to consider re-submission.

We now better quantify the role of Indian Ocean SSTs for the water vapor response to ENSO, as we noted in the original submission that this feature was remarkable in the 97/98 event. Specifically three new figures have been added. This new analysis clarifies the source of the nonlinearity. This analysis is copied above.

Specific comments:

p.6, l.14-15: "EN leads to strong cooling" will be OK, but "LN leads to warming" is not obvious since the vertical axis is anomalies.

We now clarify "LN leads to warming relative to the climatology"

p.6, l.29: "This is especially evident in Figure 1il": In Figure 1i, the negative value of slope appears statistically significant, which may be interpreted as a linear response.

In response to reviewer 1, we have modified the way we display nonlinearity. As the slope is no longer displayed on this panel, this comment is not relevant in the revised manuscript.

To answer the reviewer more directly, the utility of a linear fit can be quantified using R^2 and adjusted R^2, and in this case the adjusted R^2 of a polynomial fit is substantially higher the R^2 of the linear fit.

Figure 4. The choice of green (+) and red (-) in color scale is confusing; the choice of the same color as in other figures (Figs. 5, 6 and so on) is recommended.

We agree that our original scale was confusing, and we now adapt a scale that we have seen used for precipitation in e.g. Kang and Polvani 2011. An example is below.

[Figure]

Figures 5 and 6: The distribution of cold region is only one aspect of TTL dehydration. There is no information on the location of the dehydration that is taking place for the air entering the stratosphere. The distribution of Lagrangian cold point was reported to have changed dramatically during 1997/98 El Nino (Fig. 8 of Hasebe and Noguchi, 2016).

As discussed above, we are unable to run a full trajectory model. However, the shift in the LCP in Hasebe and Noguchi appears consistent with the temperature changes shown in our figure 6. We now note this similarity and cite Hasebe and Noguchi in the text.

p.10, l.5-6: "regardless of their type": It seems the nonlinearity appears only in EP type El Nino (2015/16 is also categorized as EP EN).

These words have been removed

Reviewer #2 (S. Fueglistaler)

Garfinkel et al. study the effect of ENSO on temperature around the tropical tropopause, and related to that on water entering the stratosphere. They study the problem using observations of stratospheric water vapor (the SWOOSH data set), tem-peratures from the MERRA reanalysis, and climate model simulations in a range of configurations. The paper's main point is that the response in tropopause temperature and water entering the stratosphere is non-linearly related to ENSO (as represented by some index) - such that both strong El Nino and La Nina lead to a temperature increase (and correspondingly moistening of the stratosphere). Using this result, they argue that the sequence of strong El Nino followed by strong La Ninas in the late 1990's led to elevated temperatures (and moister air) for a few years, which contributes to the 'drop' observed around September/October 2000. The hypothesis put forward is very interesting - but I have a number of concerns.

I could not help the impression that this paper was written up slightly careless. At times, the text reads more like a story than a scientific paper; similarly, the paper has problems finding the right tone. Consider the abstract. There, it is first written that: "The impact... is nonlinear." which leaves no room for doubt. However, the next sentence does not provide the hard evidence expected by the reader, but uses the rather weak word "appear" twice. Also, considering the seemingly straightforward hypothesis - the non-linearity of ENSO - I expected to be shown a plot that shows the non-linearity beyond doubt. Instead, the paper presents a full 16 figures that show a lot of information - most of it only qualitatively discussed. The paper would be much stronger if the authors were able to support the main point of their paper in one or two clearly drafted figures.

We agree that our original submission was a bit vague and qualitative. We have also removed three of the original figures as the discussion of them was (admittedly) overly hand-wavy, and they weren't truly necessary. Five other figures have been moved to supplemental material. The language in the abstract has been modified as well to be more confident. Finally, we have added several new figures that nail home the importance of Indian Ocean SSTs in a more quantitative way, as this is indeed the source of the nonlinearities. The revised version adopts a more scientific tone and hopefully is more convincing.

Figure 1 presumably presents the model data that best supports the argument for non-linearity-however the points are so small, and cyan has very little contrast, such that it is easy to overlook the datapoints supporting the hypothesis. Simply tweaking colors and symbol size would probably help.

Figure 1, coupled with the response to 97/98 shown in figure 2 (figure 2 and 3 in the revised manuscript), are indeed the figures that show the nonlinearity most clearly.

The marker sizes and colors have been changed in order to enhance readability. See figure 10 above (the third of the figures that relates to the importance of Indian Ocean SSTs) as an example.

Also, I'd like to see a more quantitative treatment of the non-linearity (i.e. it would be simply to compare the linear regression with a non-linear regression).

We now utilize the adjusted R-squared test, a standard technique in many disciplines to quantify the relative goodness of fits for a simple linear test and for a parabolic test.

Specifically, for all panels in figures 2,3, 4, 11, 13-16 we compute the adjusted R-squared using a polynomial fit and compare it to the R-squared for a simple linear fit. When the polynomial adjusted R-squared is more than 4/3 the linear R-squared, we adopt the polynomial fit. Note that in principle the 4/3 factor isn't necessary – if the adjusted R-squared is any higher than the linear R-squared then the polynomial fit is to be preferred. However we elect to be somewhat conservative and avoid over-fitting. For panels in which a polynomial fit is preferred, we show the adjusted R-squared for the polynomial fit and the linear R-squared. An example is in figure 10 above (the third of the figures that relates to the importance of Indian Ocean SSTs).

In all cases, the parabolic fit is better when we intuited it would be better in the original submission.

Also, it would be fair to show the statistical uncertainty in the "observational" data shown in Figure 3; we should be honest that the observational timeseries is really (too?) short to make statistically robust statements.

We aren't sure what exactly the reviewer intends as figure 3 (now figure 4) included a best fit line with 95% uncertainties. However we now note explicitly in the text that none of the regression lines for water vapor in SWOOSH are statistically significant.

The paper then applies the argument of non-linearity to explain the sudden and persistent drop of stratospheric water vapor around October 2000. However, there is a major conundrum pointed out in Fueglistaler et al. (J. Geophys Res., 2013) that needs to be addressed here: The arguably best representation of true temperatures in re-analysis data fails to properly produce a drop as observed in HALOE data. That is, the mechanism discussed in this paper applies to the large-scale effect of temperature and circulation, but the problem is that even if the free-running GCM would recover the reanalysis temperature perfectly, it would not be able to produce a drop as prominent as observed by HALOE. Correspondingly, it is not surprising that the drop diagnosed by the authors is only 23% of the HALOE drop. While there is plenty of good reason to have trust in HALOE data, it is crucial to note here that the stratospheric water vapor time series as observed by SAGEII agrees very well with the reanalysis-based model estimates (Fueglistaler et al., 2013). This

needs to be discussed; and I would strongly encourage you to also consider quantifying the importance not against HALOE, but against the AMIP-mode model generated data (this helps your paper). However, the analysis of the drop as presented in Figure 11 is close to cherry-picking: anyone can see that what is labelled here as "decadal drop" is anything but a decadal drop. I also suspect that this time series does not compare favourably against SWOOSH at all - should this not be reason for serious concern given that this is an AMIP run?

We have significantly shortened the discussion of the water vapor millennium drop, as this is more of a secondary point of this paper than the main point. We also unfortunately set up unrealistic expectations as to what an AMIP model can actually accomplish as to capturing observed water vapor variability. We now clarify our expectations for the ability GEOSCCM to capture observed water vapor variability at the beginning of section 6. Specifically, we note that the QBO and BDC in GEOSCCM does not match that in observations, and hence there should not be any specific expectation that the specific timing and magnitude of the drop should match that observed.

Rather, these experiments are useful for one purpose: quantifying the contribution of SSTs to the drop in 2000, and we limit our discussion to this point in the first half of section 6 (i.e. before we move onto 2015/2016). That the contribution is relatively minor (approximately 23%) is consistent with previous work and the reviewer's intuition that other forcing factors (such as the BDC and QBO, or perhaps phenomena unrelated to cold point temperatures) are more important for entry water vapor variability near the end of 2000. However other studies have suggested that SSTs play a role, but the literature lacks a clear quantification of their role.

Quantifying the magnitude of the drop from a given satellite product is difficult due to missing data, and deciding among the many different ways to account for this missing data is beyond the scope of this paper. However, we agree that it important to note that there are discrepancies. Specifically, we have added that "the different satellite products underlying SWOOSH disagree as to the magnitude of the drop", and cite Fueglistaler et al 2013. We are now careful to write that **approximately** 23%  of the observed drop is associated with SSTs, as the reviewer is correct that we do not know precisely how big the drop actually was.

To summarize, the paper by Garfinkel et al. touches many interesting points, and makes use of interesting numerical model runs. The paper needs, however, a major overhaul; the main points need to be worked out clearer in the data, and the discussion of the "drop" needs to be more careful. Given the recommendation for major revisions, I do not go further into the details of the current manuscript.

---

## Author Response (AR2)

**Review by  Stephan Fueglistaler:**

Review of revised version of "Nonlinear response of tropical lower stratospheric temperature and water vapor to ENSO"

by Garfinkel et al.

For the revised version, Garfinkel et al. have addressed many of the concerns raised by reviewers. In particular, the revised version is more focused, the main points are clearer, and limitations are discussed. Some minor comments are listed below that can be addressed easily (the list below is not comprehensive, also, generally the text would deserve some polishing).

After reading the revised manuscript multiple times, my understanding is that the essence of the paper is as follows: the free running GCM produces statistically signigicant, and visually compelling non-linear relations between ENSO index and lower stratospheric properties (temperature, water vapor). The AGCM runs, forced with SSTs of the last four decades, produce at least some non-linear response - the weaker statistics are presumably because the last four decades only had one or two sufficiently strong (in terms of ENSO index) El Ninos, whereas the free running GCM calculations have a sufficiently large number of very strong ENSOs, as seen in Figure 2. Finally, because the AMIP GCM calculations can be run many times (ensembles), their response is statistically more robust than what was actually measured (i.e. the non-linearity in the observational record as shown in Figure 4 hinges essentially on one data point). While this chain of arguments requires quite a leap of faith (when looking at the observations displayed in Figure 4 in isolation, the notion of non-linearity is rather disturbing), I consider the results from the coupled ocean-atmosphere GCM fairly convincing, and therefore recommend publication of the paper.

Because of the paramount importance of the coupled ocean-atmosphere model calculation for the central argument of the paper, I think the authors should address the question whether the GEOSCCM coupled to MOM5 produces an ENSO sufficiently similar to reality. In this respect, the fact that it produces stronger ENSOs than observed as noted on P4/L24 is not comforting - but this could also be an artifact related the definition of the NINO3.4 index, and GEOSCCM-MOM5 having (most likely) a bias in the mean state. For the revised version, I'd encourage the authors to provide more information about ENSO biases in their model, and reassurance that they do not critically affect the postulated non-linearity.

Li et al 2016 document mean state biases in the GEOSCCM-MOM5 system. SSTs are biased warm in the west and central Pacific in GEOSCCM-MOM5 – see figure 4 of Li et al 2016 – consistent with zonal wind stresses that are insufficiently easterly (figure 3 of Li et al 2016). Similar biases appear to be present in GFDL's CM2.6, to our knowledge the last ocean-atmosphere model from the GFDL before the upgrade to MOM6 (see https://www.gfdl.noaa.gov/wp-content/uploads/2017/12/2-1_Winton.pdf). We now note that Li et al 2016 document these mean state biases.

There is no paper that documents the ENSO properties in GEOSCCM-MOM5. Figure R1 below shows the SST anomalies in December and January for events in which the Nino3.4 index falls between 0.5 and 1.5K (top row), 1.5K and 2.5K (middle row), and 2.5K and 3.5K (bottom row). In all cases SST anomalies are present in the Pacific (indeed by construction). SST anomalies are also present in the Indian Ocean, and as shown in Figure 1 in the paper this feature is also realistic. The ENSO simulated by the model appears to be realistic except that it is too-vigorous.

Note that the GFDL earth system model, which contains the previous iteration of the ocean model we use, also has a too-large ENSO amplitude (Copotondi et al 2015; https://extranet.gfdl.noaa.gov/~atw/yr/2015/capotondi_etal_variations2015.pdf and Dunne et al 2012). The commonality of this bias suggests that this ocean model may have issues with ENSO amplitude. We now mention Dunne et al and Copotondi et al in the methods/data section. We have also added a sentence to the final paragraph of the conclusion noting that the ENSO amplitude in our coupled model is too large (in addition to the sentence that was already included in the data/methods section).

[Figure]

El Nino, Nino3.4 between 0.5 and 1.5K;  40 events

[Figure]

El Nino, Nino3.4 between 1.5 and 2.5K;  24 events

[Figure]

K

[Figure]

El Nino, Nino3.4 between 2.5 and 3.5K;  18 events

[Figure]

**Figure R1: SST anomalies in the coupled ocean-atmosphere- GEOSCCM simulations in December and January for events in which the Nino3.4 index falls between 0.5 and 1.5K (top row), 1.5K and 2.5K (middle row), and 2.5K and 3.5K (bottom row). The number of events included for each composite is listed. The zero line is indicated in black.**

Minor comments/typos:

P2/L19: "moreso"

**"moreso" has been changed to "more"**

P2/L25: "strengthed"

**We now write "comparable anomalies"**

P4/L31: Observed SSTs: It is worth noting that some care is required regarding "observed" SSTs as these datasets are subject to similar problems as the atmospheric temperature record; and right around the first strong ENSO event 1982/83, for example, the HURRELL and HADSST1 dataset diverge substantial in their tropical mean (See Figure 1c, Flannaghan et al., JGR, doi:10.1002/2014JD022365., 2014). Given that the Indian Ocean signal is order 0.5K, it would be worthwhile to check whether this signal is consistent among different SST datasets.

**Figure 1 in the manuscript is based on ERSSTv5, and is indistinguishable from a similarly constructed figure using ERSSTv4. We have also computed the relative size of anomalies in the Indo-Pacific (50-150E, 5S-5N as in figure 1 and 10) for 97/98 and 82/83 from HadISST and the anomaly in spring of 1983 is 0.3K, while in spring of 1998 it is 0.6K. This is consistent with figure 1. We now note this explicitly in the paper. We have also confirmed that other EN events that occurred after the time period discussed in this paper are associated with warming in the Indo-pacific (e.g. 2009/2010) in both dataset.**

P7/L8: Add "NH" or "boreal" before "winter".

**"boreal" has been added**

P7/L24: "tha*t* might ..."

**"that" has been changed to "than"**

Figure 5: caption - if I am not mistaken this is an AMIP run, please label as in text (P8L8) "AGCM GEOSCCM"

**"AGCM " has been added**

**Co-Editor Decision: Reconsider after major revisions (21 Jan 2018) by Peter Haynes**

Comments to the Author:

Referees 1 and 2 were both strongly critical of the first version of this paper.

Referee 2 (Stephan Fueglistaler) has now considered the revised paper and his recommendation is that the paper now be accepted after minor revision. He sets out various detailed comments.

Referee 1, who recommended rejection of the paper in the first instance, does not have time for a detailed review of the revised version of the paper. Therefore I have decided to provide such a review myself, focusing on the points that were of particular concern to the Referee.

Referee 1 originally raised the following points:

1. Analysis is limited to temperature response and does not consider any effect of variation of pathways.

2. The discussion of the observed millennium drop is imprecise — for example there is vagueness about timing of a drop seen in simulations vs the timing see in the observational record.

3. Various terms are not precisely defined.

4. Greater precision is needed in the discussion of 'nonlinearity' and its possible relation to the 1997/98 El Nino event. Arguments for the importance of Indian Ocean surface temperature anomalies in the nonlinearity are poorly supported by evidence.

5. The discussion of the detailed predicted time variation of water vapour is poorly related to the observed record — for example there is a predicted significant drop in water vapour in 1997 but that is ignored in the discussion.

6. The reason for considering mean age is not made clear and the method for calculating mean age is not properly described.

Referee 1 also made a number of detailed comments, but emphasised that they had not provided an exhaustive list of recommended technical corrections because their overall recommendation was that the paper required a complete rewrite.

Having looked at your revised paper, my impression of your response to the above comments is as follows.

1. You have acknowledged that there is no consideration of the role of changing pathways in the determination of the water vapour response to El Nino, but in a rather indirect way which seems to suggest that variation in pathways might be responsible for variability between ensemble members rather than for systematic variation. My recommendation is that you make this acknowledgement more conspicuous and, unless you can justify the argument carefully, you remove the suggestion that variation in transport pathways is somehow responsible for variability and not for systematic variation.

**We agree that the temperature response to El Nino also differs among ensemble members – indeed we show this explicitly in Figure 3 – and hence differences in the temperature response also lead to diversity in the model-simulated entry water vapor. We also agree that "sampling variability" could lead to systematic differences and not just intra-ensemble diversity. Our comments in the response to the reviewers was intended as speculation, and this speculation in the response to the reviewers was not included in the revised text except in the conclusions/discussion. Hence we have removed this comment from the conclusions. This point as discussed below in greater detail.**

2./5. Your discussion of the relevance of your results to the millennium drop needs to be further clarified.

3./6. have been dealt with to some extent.

4. I think that there has been some clarification here, but there could be further improvement.

Having looked at the paper myself, and taking into account Referee 2's comments, I see the paper as now in a state of potentially publishable if revised further.

I have set out a list of detailed comments below. Please consider these in addition to those from Referee 2 and provide a revised version of the manuscript, plus responses as appropriate that address my comments and those of Referee 2.

I hope to accept the paper after further revision and without further consultation with referees.

DETAILED COMMENTS:

p1 l20: 'these regions' > 'these two regions'/'the polar and the tropical regions'

**changed to "these two regions"**

p1 l21: 'During an EN event' — of course all this is subject to internal dynamical variability — see for example Hardiman et al 2008.

**We agree that this response is not present in all events and can be masked by internal variability. (We are currently writing a follow-up paper that addresses the signal-to-noise ratio more explicitly.) We have changed to "During most EN events, …"**

p2 l19: 'moreso' > 'more'?

**changed**

p2 'led to 0.14ppmv of dehydration, explaining approximately 23% of the observed drop in water vapor over this period. 'explaining approximately 23% of the observed drop' seems a slightly absurd statement. (There is a similar statement in the abstract.) Surely 'around one-quarter' would be a much better way of saying this. Also at some point in the paper you need to say carefully what you mean by this — you are deducing from a large ensemble of AGCM simulations with imposed SSTs that the deterministic part of water-vapour drop arising from these imposed SSTs is about one-quarter of that actually observed.

**Yes, this is indeed absurd. We have changed to "one-quarter" throughout the manuscript. We have included a sentence very similar to your suggestion here.**

**"Finally, by comparing changes in water vapor concentrations between the early 2000s and late 1990s in a large ensemble of model simulations forced with observed sea surface temperatures, we suggest that the deterministic component of the water vapor drop in the early 2000s was  0.14ppmv, approximately one-quarter of the observed drop. "**

p2 l33: 'such that ENSO variability aliased onto sub-decadal variability' — I suppose that if one thinks of analysis of interannual variability as a kind of black-box exercise in time-series analysis then this comment makes some kind of sense. But my suggestion is omit it.

**We have removed this comment. (Our original intention is that the time-spectrum of ENSO variability has a tail that extends beyond 7 years, but there is no reason to belabor such a minor point)**

p3 l4: It seems to me that in the end you are seriously questioning the use of linear regression only with respect to water vapour (and you don't really have any grounds for questioning its use beyond that). You need to make that clear.

**It is also problematic for temperature, as is evident in figures 2 and 4.**

**The title of our paper only claims nonlinearity for temperature and water vapor. However there were several instances in the text where we weren't sufficiently explicit that nonlineatiy only applies for these two variables, and we have added the qualifier that nonlinearity is present for temperature and water vapor only wherever relevant, and also that the nonlinearity is only evident in spring and early summer.**

p3 l19: 'The source of this nonlinearity is the Indian Ocean response to EN' — this is a pretty strong statement — you are providing some evidence for this. Make it clear that this is a suggestion, not a fact.

**Our original statement was perhaps a bit too strong, but "suggestion" seems too weak. We now write "This nonlinearity apparently originates in the Indo-West Pacific response to EN".**

Figure 1: This is labelled 'relation between Indian Ocean and ENSO near surface temperatures' but the field being considered is 50E-150E temperature, which you refer to you yourself as 'Indo-Pacific'. Only about half the 50E-150E region is the Indian Ocean. Clarify this.

p3 l24: 'The tendency of EN events to lead to a warmer Indian Ocean is well captured by the model.' As noted in previous comment this can't be deduced from what is shown given that only about half of 50E-150E is Indian Ocean. Amend.

**We have added two additional panels to figure 1 that focus only on 50E-100E. Results are mostly indistinguishable and if anything slightly stronger.  (Our original choice of 50-150E was motivated by figure 9, as discussed later). We have clarified Indian to Indo-West Pacific throughout the manuscript.**

p5 l15: 'Model output necessary to run a Lagrangian trajectory model for these simulations was not archived and hence we cannot quantify the specific location of dehydration.' Something like this comment was requested by Referee 1 and is needed somewhere but it seems out of place here. It is really an analysis/interpretation issue rather than a model issue. The important general point is surely something like 'Full explanation of inter annual variability in stratospheric water vapour, particularly that associated with El Nino (Bonazzola and Haynes 2004, Hasebe and Noguchi 2016, Konopka et al 2016), requires consideration of both a 'sampling effect' and a 'temperature effect'. These two effects cannot be distinguished in our simulations, since the model output necessary to run a Lagrangian trajectory model was not archived. Nonetheless neither is prevented from operating in the simulations and the simulated interannual variability in water vapour will arise from some combination of the two.' — that needs to be said somewhere in the early part of the paper and I suggest (given that Referee 1 raised this as an important point) that it is also repeated somewhere in the conclusions.

**We have replaced the sentence here with a slightly modified version of what you suggest. We have also moved this paragraph and the one that follows to the introduction. The revised  text is copied below:**

**"A complete explanation of inter-annual variability in stratospheric water vapor, particularly that associated with El Nino (Bonazzola and Haynes 2004; Hasebe and Noguchi 2016; Konopka et al 2016), requires consideration of changes in both temperature and air-parcel trajectories near the tropopause, and might also be influenced by changes in cloud ice (Avery et al 2017). We cannot distinguish among these various effects in our simulations, since the model output necessary to run a Lagrangian trajectory model was not archived. Nonetheless all of these effects operate in the simulations, and the simulated interannual variability in water vapor will arise from some combination of these effects**. "

p5 l30: 'ENSO events in the simulations'?

**This categorization is used both for reanalysis/SWOOSH and the model simulations. This sentence has been clarified.**

p6 l10-13: I found these sentences very difficult to understand. For example by 'moistening of the stratosphere during El Nino is more pronounced that expected' you mean something like 'the El Nino moistening signal is stronger if the BDC is not regressed out of the water vapour signal' but I can't tell exactly what. Terms like 'more pronounced than expected' often cause problems because it might be what you expect but it might not be what someone else expects.

**We have removed the sentence from line 12 to 13, and rewritten the sentence from line 10 to 12. The revised version is copied below:**

**" As discussed in the introduction it is well known that EN forces an intensified BDC, and associated with an accelerated BDC are colder tropical lower stratospheric temperatures and less water vapor. Here we consider the response to ENSO without regressing out the influence of the BDC on water vapor except where indicated, as regressing out the BDC misrepresents the net impact of ENSO on the lower stratosphere"**

p6 l17-21: This paragraph is also not easy to understand. 'many of the complications that arise due to the QBO (e.g. Liang el al 2011) are not relevant' is I believe your particular take on the fact that there is a strong QBO effect on e.g. tracer distributions in this region — which are the subject of the Liang et al paper — but that in your ensemble of simulations there is no time coherence of the QBO across the ensemble — so when you consider any kind of ensemble average then you do not expect to see any coherent QBO signal. However you do apparently take account of the QBO when you compare against observations — though I don't understand exactly what you mean. Do you mean that the QBO signal is taken out of the observations? or

out of the simulation results? This is further confusing because you seem to have 'removed' the QBO in Figure 2 and 'detrended QBO regressed', whatever that means in Figure 3 — but neither of those individual Figures seems to involve 'comparison against observations'. Clarification is definitely needed.

**We have clarified this paragraph, and the revised version is copied below:**

**"A QBO is spontaneously generated in all simulations considered here. The QBO phase is not coherent among these experiments (i.e. the phase does not match observations), and hence the impact of the QBO on e.g. tracer distribution (Liang et al 2011) is averaged out when considering the ensemble mean. As the QBO does impact tracer distribution in observations, however,}we linearly regress out variability associated with the zonal wind at 50hPa two months prior before considering the response to ENSO."**

p6 l24: 'on Figures of temperature at 100hPa' — telling the reader exactly which figures would be helpful.

**corrected**

p7 l3: 'from the late fall through late spring' — both of these are NH of course (as are presumably any references to season).

**Changed to "November through June", and also clarified throughout.**

Figure 2: The markers are very small. I found it almost impossible to distinguish between blue and black, for example. Please modify. The letters and numbers in the Figure annotations/extra information are also very difficult to read. Sometimes these numbers, e.g. R^2 values, are important. Please modify.

**The blue and black markers have been enlarged, as has the text in gray for the R^2 values and best-fit slopes.**

Figure 3: Are (d),(e),(f) correctly labelled — currently as 'Mar-Jun'? You might note explicitly somewhere the very large role of internal variability as manifested by the very large spread in values across the ensemble. Actually there is presumably no particular difference in this between the AGCM simulations and the coupled simulations — it is just that with the coupled simulations the number of ensemble members is much smaller so pattern of spread in values is less 'full'.

**They were indeed incorrectly labeled – it should have been May-Jun. Corrected.**

**We have added a sentence about the importance of internal variability for the spread in response for a given event.**

Figure 3 caption: 'integration' is presumably the same as 'ensemble member'.

**Corrected.**

p7 l16: 'Even if we linearly regress out the BDC the slope … (not shown).' Given that the results can't be seen is it really important to make this point here?

**This sentence has been removed.**

p8 l2: Is it really fair to describe Figure 4 as 'the observational constraints' or indeed to put any significance that the simulations fall within such 'constraints' (in a sense that you haven't precisely defined). From my point of view the single 30-year or so time series of the observations might be expected to fall within the envelope mapped out by a large ensemble or such time series, not necessarily the other way round. As you seem to say, the 'nonlinearity' in Figure 4 seems to come down to the presence of the year 1997/98 — the extra information added by the nonlinear curve fitting — with a small value of R^2 — is not at all clear.

**We have removed the remark about observational constraints, and revised this paragraph. The revised text is copied below:**

**"The response to ENSO in GEOSCCM can be used to inform the interpretation of the observed response to ENSO  (Figure 4).  EN leads to an accelerated BDC and a colder lower stratosphere in reanalysis data in January and February, and these changes are statistically indistinguishable from the response in GEOSCCM. More importantly, the qualitatively different behavior for the 1997/1998 event as compared to moderate EN events in the model experiments is also evident in observations in March through June, and hence we recommend caution in generalizing about the tropical lower stratospheric temperature and water vapor response to EN events from the observed anomalies in 1997/1998. } However, the relatively short data record limits the confidence with which we can identify nonlinearities in observational/reanalysis data,   and none of the  linear best-fit slope estimates for SWOOSH water vapor    are statistically significant in either winter or spring.  "**

**We have elected to show the R^2 for the observational figure in order to maintain the same methodology for all figures,  though if the editor insists we can switch to linear best-fits for all panels on this figure.**

Figure 5: I'm assuming that in this figure some effect of the QBO signal has been 'taken out' (whatever that means exactly). It might actually be most useful to postpone proper explanation of this until this point and then explain it clearly.

**We now refer to the methods section where this procedure is described in more detail.**

p8 l11: '0.4 ppmv in late (NH) spring' — the value of 0.4 ppmv looks to me to start only in June — isn't that NH summer, not NH spring.

**Changed from "spring" to "June"**

p8 l12-23: This discussion in the difference in the shape and location of cold point regions in 97/98 versus other years is quite difficult to follow and I'm not sure that your addition of particular coloured temperature contours has succeeded in making things clear. But if you cannot see a better way to do things then so be it.

**We have revised this paragraph to improve clarity, and the revised text is copied below:**

**"Figure 6 and 7 show a map view of changes in temperature at 100hPa for the 97/98 event and for all other EP EN events. The green contour on each panel surrounds the coldest region of the Tropics climatologically, while the magenta contour surrounds the coldest region of the Tropics during the specific EN composite. In both Figure 6 and 7 there is relative cooling between 170W and 120W and relative warming over the Warm Pool region from November through February, but the longitude of the nodal line between warming and cooling differs between the 97/98 event and all other EP EN events. Specifically, in the 97/98 event in boreal winter, the zero-line of temperature anomalies is 30degrees further east than for the other EP EN events (compare the black zero-line in Figure 6b and Figure 7b), such that during the 97/98 event the entirety of the climatological cold point region warms. The net effect of this warming of the climatological cold point region is that the cold point shifts to the east while warming during 97/98 (the magenta isotherm is 0.7K warmer than the green contour in Figure 6). In contrast, during other EP EN events, roughly half of the climatological cold point region warms while the other half cools, and the net effect is that the coldest region shifts east but does not warm or cool overall for typical EP EN events (the green and magenta isotherms in Figure 7 correspond to the same temperature). The eastward shift in Figure 6b and 7ab is consistent with the shift in the Lagrangian cold point evident in figure 8 of Hasebe and Noguchi 2016. In boreal spring, there is broad-scale warming over most of the equatorial band for the 97/98 event (Figure 6cd), while the temperature anomalies are similar to those in winter for moderate EN events (Figure 7cd). A similar effect is seen in the MERRA reanalysis (not shown). The net effect is that in boreal winter and especially spring, the 97/98 event led to warming of the cold point and moistening of the stratosphere relative to other EP EN events. "**

p8 l30: 'this nonlinearity in the temperature and water vapour response appears to originate from the troposphere' — isn't the most compelling reasoning for this simply that in the AGCM simulations (which you are considering) the 'input' signal is the SST field. But what do you mean by 'originate from the troposphere' exactly? Do you mean 'does not involve stratospheric dynamics'?

**Changed to "does not involve stratospheric dynamics"**

p9 l13: I've already noted in a previous comment then if you are particularly emphasising the role of the Indian Ocean then using an index based on the 'Indo-Pacific region' 50E-150E, half of which is not in the Indian Ocean, seems not the best choice.

**Figure 9 suggests that it is more than just the Indian Ocean that matters for water vapor; rather the entire Indo-West Pacific region is important. We now clarify this point, and specifically have replaced Indian with Indo-West Pacific throughout.**

p9 l17: You show curves regressing out the BDC and linear regressing out the QBO. So do I conclude that the QBO signal is retained in the curve which regresses out the BDC. Why?

**We did indeed originally not include the QBO regression, but now do. Results are indistinguishable.**

The importance of these regression coefficients depends of course in part on the magnitude of the temperature variations in the different regions. The coefficient for a particular region could be large but the magnitude of the temperature variation could be small. You should comment on that.

**We now note that the importance of a large regression coefficient in a given region depends on the magnitude of near-surface temperature variations in that region.**

p9 l23: 'In the annual average, warmer near-surface temperatures over the Central and Eastern Pacific lead to dehydration of the stratosphere in all three data sources' — for clarity, what features exactly in Figure 9 are you identifying that prompt you to say this?

**We now refer to the black curve in figure 9ace.**

Section 6: Referee 1 asked for more clarity on what you mean by the millennium drop and you have made some modifications, but there still seems to be scope for further clarity. I suggest that you move and modify your sentence p10 l23-25 to close to the beginning of the section. Then it is clear to the reader what you mean by the drop and also what aspects of the drop — e.g. the detailed time evolution of the late 2000 and early 2001 period.

**We have added to the first paragraph of section 6 that we are referring to the difference between 2002 to 2004 versus 1998 to 2000. We have also better motivated why ENSO would matter for water vapor evolution in this period. The revised text is copied below:**

**"Before proceeding, it is important to mention that the 1997/1998 El Nino was followed by nearly three consecutive years of strong La Nina conditions - the Nino3.4 index in the ERSST5 dataset did not drop below -0.5K until March 2001 - which was then followed by weak El Nino conditions from 2002 through 2004. As discussed above, strong La Nina events also lead to moistening of the stratosphere, while weak El Nino  lead to dehydration. The net effect is that ENSO was in a phase that leads to enhanced water vapor during 1998, 1999, and 2000 and in a phase that leads to reduced water vapor from 2002 to 2004. ……. These experiments can be used to quantify the contribution of SSTs to the difference in water vapor between 2002 through 2004 and 1998 through 2000."**

p10 l31: 'Hence SST changes contributed to the drop … but were not the major forcing factor …' — these seem to be overconfident conclusions. You need to say something more measured — e.g. 'on the basis of your large ensemble of AGCM simulations, the imposed SST signal can account for a ensemble average water vapour drop (by your definition) of about 0.14pmmv. This suggests that … '.

**We have added "our GEOSCCM simulations suggest that" to this sentence, and earlier in the paragraph we have added "the imposed SST signal can account for an ensemble-averaged dehydration of about 0.14ppmv"**

p10 l32: 'consistent with Garfinkel et al (2013b)' — actually in this paper you say nothing at all about 'the drop' as far as I can tell. So I don't understand why you are referencing it. In the reference list you also have the title (of your own paper!) listed incorrectly — it doesn't match the title of the paper that appeared. For that matter Garfinkel et al (2013) is presumably no longer 'in press'.

**We apologize for the mistake – Garfinkel et al 2013 (on EP vs CP ENSO) was indeed published several years ago.**

**On this line we  were referring not to the Garfinkel et al 2013 paper on CP vs EP El Nino and water vapor, but rather to a different one on zonal asymmetries in the TTL and SST trends. The title for that paper was indeed listed incorrectly, and we apologize for any confusion. This paper did indeed discuss the "drop" –see the appendix.**

p10 l32-33: 'The magnitude of the drop is 0.09ppmv if we consider water vapour area weighted form 60S to 60N.' I don't understand why this sentence has been included. It doesn't seem to reference anything else. Omit?

**removed**

p11 Figure 11bc: To me these figures on cloud ice is a distraction. No detailed discussion is given. This is the first time that cloud ice has been mentioned at all in the paper. Figure 11c shows December and increased cloud ice over the Central Pacific when in the rest of the paper you seem to have been emphasising that moistening occurs in 'NH spring' and is more associated with processes over the Indo-Pacific region etc. There seem to be increases of up to 2ppmv at 85hPa which doesn't fit with the statement of 'even at 85hPa cloud ice increases by 0.05ppmv'. I recommend to postpone any discussion of cloud ice to a further publication where details can be presented properly.

**We have moved the discussion of cloud ice to the conclusions/discussion section in the paragraph that discusses "unanswered questions", where we explicitly acknowledge that future work must be performed to better understand the pathways whereby ENSO modulates**

stratospheric water vapor. However the Avery et al paper highlights that cloud ice may be important for the stratospheric water vapor response to ENSO, and hence we believe it is important to show that the model can capture this effect at least qualitatively.

In the zonal mean cloud ice at 85hPa increases by 0.05ppmv. Locally the anomalies are 2ppmv. This has been clarified

p11 ll16-17: 'Hence in summary, strong EN events lead …' — OK, but it is slightly confusing the previous sentence is completely unrelated to this.

The discussion of cloud ice and of the changes in 2011 have been moved, such that this concluding sentence more naturally follows the discussion of the 2015/2016 event.

p12 l4-8: Again this conclusion confuses 'truth' with a useful line of argument deduced from an important set of model simulations. We don't actually know if the 'enhanced' water vapour in 1998-2000 was due to El Nino/La Nina or not. What we do know from your simulations is that providing observed SSTs can lead (in the sense that it leads to an ensemble average signal) decrease in water vapour from 1998-2000 to 2002-2004 of 0.14ppmv, so it that sense some of the observed drop may well have been 'caused' directly by the SSTs. You say that this accounts for 'approximately 23%' of the observed drop. I might say this shows that ONLY about one quarter of the drop can be directly accounted for by the SSTs.

We have modified this paragraph as follows:

The very strong El Nino event in 1997/1998 followed by more than two consecutive years of La Nina led to enhanced lower stratospheric water vapor. As this period ended in early 2001, entry water vapor concentrations declined. We quantify this effect using a large ensemble of AGCM simulations with imposed SSTs, and find that the deterministic part of the water-vapor drop arising from these imposed SSTs is about one-quarter of that actually observed, in agreement with the recent estimate of Ding and Fu (2017) who used a different model.  Hence, it is important to consider SST variability when considering decadal variability in the lower stratosphere, though other forcings were more important for the millennium drop  as only one-quarter of the drop can be directly accounted for by SSTs.

p12 l21: 'However these sampling effects are included implicitly in GEOSCCM' — that is fair comment. (See other comments above.) 'some of the diversity in response among the 42 ensemble members to an identical SST forcing is almost certainly due to such sampling effects' — I don't understand why this statement is being made and it might be revealing a confusion. Ensemble members with the same SST forcing will differ in both TTL circulation and TTL temperature and in that sense both 'sampling effects' and 'temperature effects' will lead to differences between such ensemble members. There is no sense in which 'sampling effects' are more about inter-ensemble variability and 'temperature effects' are more about ensemble mean response. The ensemble mean response includes both temperature effects and sampling effects. The inter-ensemble variability includes both temperature effects and sampling effects. Again it seems to me that you simply need to be candid and say that the effects you are identifying are likely to have a contribution from temperature effects and a contribution from sampling effects and in the absence of further information you can't distinguish between the two and can't say anything more.

**We have removed the remark about the diversity in response among the 42 members, as you are correct that some of this diversity arises due to temperature effects. The wording you suggested earlier has been incorporated here, and this text now reads:**

**"Third, and relatedly, we cannot provide a complete explanation of how El Nino modulates stratospheric water vapor. Inter-annual variability in stratospheric water vapor, particularly that associated with El Nino, depends both on a 'sampling effect' (i.e. changes in the residence time in the coldest regions of the tropical tropopause layer) and a 'temperature effect' (Bonazzola and Haynes 2004; Hasebe and Noguchi 2016; Konopka et al 2016). These two effects cannot be distinguished in our simulations, since the model output necessary to run a Lagrangian trajectory model was not archived. Nonetheless neither is prevented from operating in the simulations and the simulated interannual variability in water vapor will arise from some combination of the two."**

p12 l23: You keep referring to 'the Indian Ocean' but you also refer to 'Indo-Pacific' and many of the measures that you use include longitude that are not in the Indian Ocean. It would be helpful if you could be as clear as possible on these points.

**Modified to Indo-Pacific**

[revised manuscript text omitted]

---

## Author Response (AR3)

**Co-Editor Decision: Publish subject to minor revisions (review by editor)** (17 Feb 2018) by Peter Haynes
Comments to the Author:
The paper is very close to being suitable for publication. I still have the view that you have not been sufficiently clear about the definition of post-2000 drop and what aspect of that you can address. I have previously noted that this was a significant concern of Referee 1 and I don't feel that this has yet been addressed satisfactorily.

Please address the comments below in a further revision of the paper and either make changes or provide good reasons why changes are not needed. It is the 'drop' that is my main concern -- other points are relatively minor.

Definition of millenium/millenial/post-2000 drop:

Referee 1 asked for clarity on this. Your reply re Section 6 says that you have provided this but there is still plenty of scope for improvement. At the end of the first paragraph of Section 6 you have said 'the experiments can be used to quantify the contribution of SSTs to the difference in water vapour between 2002-2004 and 1998-2000 and with these caveats duly noted we now proceed.' That's fine — you have clearly defined a drop whose causes can be assessed by your model simulations. But in the abstract you still have 'drop in water vapour in late 2000. … This … [SST] … effect accounts for approximately one-quarter of the observed drop' — which claims that you can account for a change on much shorter timescale than 2002-2004 vs 1998-2000. Please correct that particular point and re-examine your use of 'drop' across the whole paper.

p1 l10: 'millennial' — you have used 'millennial' in most places but 'millennium' in at least one place. You have used 'post-2000' in the title to Section 6. I suggest you use a single term.

**We agree with the editor that our wording in the abstract and in the section 6 title did not precisely relate to the aspect of the drop of relevance to these simulations. We instead opt for the "drop in the early 2000s" in the section title for Section 6 and in the abstract. We have elected to remove "millennial", "millennium", and "post-2000" throughout the text, as they are all overly precise for the aspect of the drop of relevance to these simulations (replaced with "drop in the early 2000s"). Otherwise, we have searched for the word "drop" throughout the manuscript, and have confirmed that changes have been made uniformly.**

p5 l3: You refer to Indo-Pacific warm pool here but actually Figure 1 also shows the Indian Ocean (50E-100E) temperatures which doesn't mentioned until a few lines later. Why not say straightforwardly that Figure 1 shows both Indian Ocean and Indo-Pacific warm pool. Also the fact that both 'Indian Ocean' and 'Indo-Pacific' are shown is not clearly stated in the caption nor in the annotation to Figure 1.

**We have changed the text to "the (top) Indian Ocean and (bottom) Indo-Pacific warm pool region". We have also clarified the caption, and the y-label on each row also now indicates the longitude range.**

p8 l25: You use 'nodal line' then 'zero line', then you refer to contours — things could be clearer.

**"nodal" has been replaced with "Zero"**

p8 l33: 'The eastward shift in Figure 6b and 7ab is consistent with the shift in the Lagrangian cold point evident in figure 8 of Hasebe and Noguchi (2016).' — good that you are referencing previous work. I can't resist commenting (without implying that any action is needed) that the same shift — again for the 1997/98 El Nino — is shown in Figures 8 and 9 of Bonazzola and Haynes (2004).

**we now reference figure 8 and 9 of Bonazzola and Haynes as well**

p10 l20: 'Strong EN events tend to have a stronger impact on the Indian Ocean than more moderate events (cf Figure 1).' — this is true in the sense that there is a positive and essentially linear correlation between an EN measure and Indian Ocean temperatures, but no more than that (e.g. no nonlinearity is evident).

[revised manuscript text omitted]